# Evaluation of Active Feature Acquisition Methods under Missing Data

## Abstract

Machine learning (ML) methods generally assume the full set of features are available at no cost. If the acquisition of a certain feature is costly at run-time, one might want to balance the acquisition cost and the predictive value of the feature for the ML task. The task of training an AI agent to decide which features are necessary to be acquired is called active feature acquisition (AFA). Current AFA methods, however, are challenged when the AFA agent has to be trained/tested with datasets that contain missing data. We formulate, for the first time, the problem of active feature acquisition performance evaluation (AFAPE) under missing data, i.e. the problem of adjusting for the inevitable missingness distribution shift between train/test time and run-time. We first propose a new causal graph, the AFA graph, that characterizes the AFAPE problem as an intervention on the environment used to train AFA agents. Here, we discuss that for handling missing data in AFAPE, the conventional approaches (off-policy policy evaluation, blocked feature acquisitions, imputation and inverse probability weighting (IPW)) often lead to biased results or are data inefficient. We then propose active feature acquisition importance sampling (AFAIS), a novel estimator that is more data efficient than IPW. We demonstrate the detrimental conclusions to which biased estimators can lead as well as the high data efficiency of AFAIS in multiple experiments using simulated and real-world data under induced MCAR, MAR and MNAR missingness.

## 1 Introduction

Machine learning methods generally assume the full set of input features is available at run-time at little to no cost. This is, however, not always the case as acquiring features may impose a significant cost. For example in medical diagnosis, the cost of feature acquisition (e.g. a biopsy test) could include both its monetary cost as well as the potential adverse harm for patients. In this case, the predictive value of a feature should be balanced against its acquisition cost. Physicians acquire certain features via biopsies, MRI scans, or lab tests, only if their diagnostic value outweighs their cost or risk. This challenge becomes more critical when physicians aim to predict a large number of diverse outcomes, each of which has different sets of informative features. Going back to the medical example, a typical emergency department (ED) is able to diagnose thousands of different diseases based on a large set of possible observations. For every new emergency patient entering ED with an unknown diagnosis, clinicians must narrow down their search for a proper diagnosis via step by step feature acquisitions. In this case an ML model designed to do prediction given the entire feature set is infeasible.

Active feature acquisition (AFA) addresses this problem by designing two AI systems: i) a so-called *AFA agent*, deciding which features must be observed, while balancing information gain vs. feature cost; ii) an ML prediction model, often a classifier, that solves the prediction task based on the acquired set of features. An AFA agent, by definition, induces missingness by selecting only a subset of features. We call this *AFA missingness* which occurs at run-time (e.g. when the AFA agent is deployed at the hospital). In addition, in many AFA applications, retrospective data which we use for model training and evaluation also contain missing entries. This is induced by a different feature acquisition process (e.g. by physicians, ordering from a wide range of diagnostic tests). We call this *retrospective missingness*. While using retrospective data (during training/evaluation), the agent can only decide among available features. At run-time, however, we make the assumption that the agent

has the freedom to choose from all features. This corresponds to a feature "availability" distribution shift that requires adjustment. Apart from difficulties of training an AFA agent using incomplete data, estimating the real world performance of agents (at run-time) using incomplete retrospective data is nontrivial and challenging. Evaluation biases might lead to false promises about the agent's performance, a serious risk especially in safety-critical applications.

This paper is the first, to our knowledge, to systematically study the evaluation of AFA agents under this distribution shift. We call this problem active feature acquisition performance evaluation (AFAPE) under missing data. Our work on the AFAPE problem contains 4 major contributions for the field of AFA:

First, we propose the *AFA graph*, a new causal graph, that characterizes the AFAPE problem under retrospective missingness as off-environment policy evaluation under an environment intervention. This is, in our opinion, a very important connection between the fields of missing data and policy evaluation (including reinforcement learning (RL)) that will result in cross-fertilization of ideas from these two areas.

Second, we show that the AFA literature only contains approaches to handle missing data that are either derived from a pure RL perspective (off-policy policy evaluation (OPE) (Chang et al., 2019), blocking of acquisition actions (Janisch et al., 2020; Yoon et al., 2018)) or correspond to biased methods for missing data (conditional mean imputation (An et al., 2022; Erion et al., 2021; Janisch et al., 2020)). We show that these approaches will almost certainly lead to biased results and/or can be extremely data inefficient. We demonstrate in experiments with exemplary missing completely at random (MCAR), missing at random (MAR), and missing not at random (MNAR) patterns that these biased evaluation methods can lead to detrimental conclusions about which AFA agent performs best. This can lead for example to high risks for patients' lives if these methods are deployed without proper evaluation.

Third, we bring the readers' attentions to unbiased estimators from the missing data literature (inverse probability weighting (IPW) (Seaman & White, 2013) and multiple imputation (MI) (Sterne et al., 2009)) which haven't been applied to AFA previously. These methods, however, do not account for the special structure of the AFAPE problem as not every feature might be acquired by the AFA agent. We show that missing data methods can, therefore, lead to data inefficiency.

Fourth, we instead propose AFAIS (active feature acquisition importance sampling), a new estimator based on the off-environment policy evaluation view. AFAIS is more data efficient than IPW, but cannot always be used for complex MNAR scenarios. For these cases, we propose a modification to AFAIS that allows it to be closer to IPW when required, at the cost of some data efficiency. We demonstrate the improved data efficiency of AFAIS over IPW in multiple experiments.

## 2 RELATED METHODS

**AFA:** Various approaches have been proposed for designing AFA agents and prediction models for active feature acquisition (AFA) (An et al., 2006; Li & Oliva, 2021a; Li et al., 2021; Chang et al., 2019; Shim et al., 2018; Yin et al., 2020). This work focuses, however, not on any particular AFA method, but on the evaluation of any AFA method under missingness. Nevertheless, we refer the interested reader to Appendix A.1 for a more detailed literature review of existing AFA methods and a distinction between AFA and other related fields.

**Missing data:** AFAPE under missingness can be viewed as a missing data problem, and hence methods from the missing data literature can be adopted. There are in general two difficulties for solving missing data problems. The first is identification, i.e. the determination whether the estimand of the full (unknown) data distribution can be estimated from the observed data distribution. The second is estimation, for which there exist generally two strategies which are based on importance sampling (IS), i.e. inverse probability weighting (IPW) (Seaman & White, 2013), and multiple imputation (MI) (Sterne et al., 2009). See Appendix A.2 for an in-depth review on missing data.

**Off-policy policy evaluation (OPE):** As we show in Section 3, the AFAPE problem can be formulated as an off-policy policy evaluation (OPE)(Dudik et al., 2011; Kallus & Uehara, 2020) problem. The goal in OPE is to evaluate the performance of a "target" policy (here the AFA policy) from data collected under a "behavior" policy (here the retrospective missingness induced by e.g. the doctor).

In this special case, the behavior policy cannot be changed, leading to offline reinforcement learning (Levine et al., 2020). Similar to missing data, there are two general OPE estimation strategies based on IS (Precup et al., 2000) and outcome regression (using q-functions (Munos & Szepesvári, 2008)). We show that these approaches are extremely data inefficient in AFA settings and can lead to biased results in complex missing data scenarios.

## 3 METHODS

We open this section by contrasting two views on missingness, which we denote as the *set view* (which focuses on which set of features was acquired), and the *sequence view* (which focuses on the order of feature acquisitions). Using the sequence view on AFA missingness, we formulate AFA as a sequential decision problem. We further introduce retrospective missingness in the set view and propose a novel graphical representation, the *AFA graph*, to describe the AFA problem under retrospective missingness. We continue by formulating the AFAPE problem as a distribution shift problem. This formulation allows us to first, discuss different views on the problem, second, study existing solutions, and finally, propose the AFAIS estimator, a novel estimator for AFAPE.

### 3.1 THE SET AND SEQUENCE VIEWS ON MISSINGNESS

The traditional view on missingness defines the problem by features $X$, missingness indicators $\bar{R}$ and observed proxies $\bar{X}$. In AFAPE, we also consider corresponding feature acquisition costs $\bar{C}$. We provide a glossary in Appendix A.13 to help the reader keep track of the different variables and terms. The relationship between variables in this view is graphically modelled by a *missing data graph* (m-graph) (Figure 1A) (Mohan et al., 2013; Shpitser et al., 2015). We name this view the *set view*, as it focuses on the set of acquired features collectively. See Appendix A.2 for an in-depth review on this view on missingness.

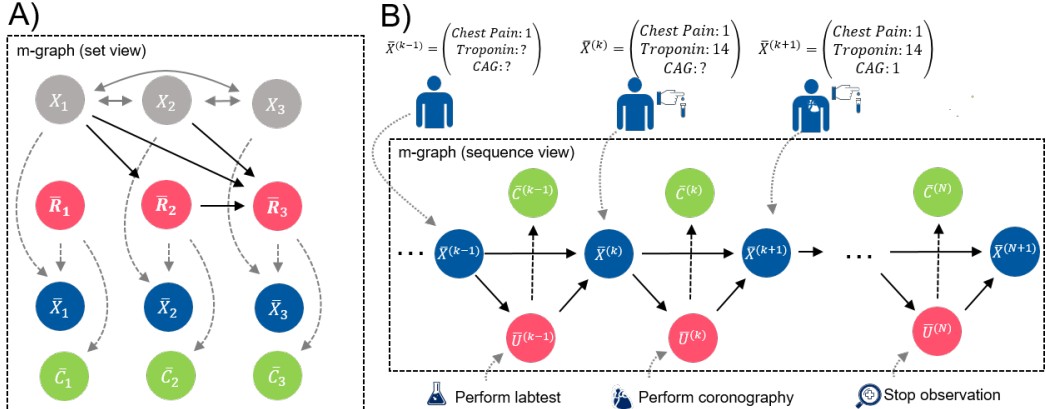

Figure 1: Missing data graphs for a simplified hospital setting of a heart attack diagnosis: $X_1$: chest pain; $X_2$: troponin (lab test); $X_3$: coronography (CAG) (invasive imaging). A) m-graph from a set view. Presence of chest pain increases the chance for the ordering of a lab test ($X_1 \rightarrow \bar{R}_2$). Based on the results from the lab test or the symptom of chest pain, a coronography might be ordered ($X_1 \rightarrow \bar{R}_3$; $X_2 \rightarrow \bar{R}_3$ ; $\bar{R}_2 \rightarrow \bar{R}_3$). The application of coronography produces a large cost $\bar{C}_3$ due to its invasiveness and radiation. B) m-graph from a sequence view, showing a (different) MAR missingness policy $\pi(\bar{U}^{(k)}|\bar{X}^{(k)}, \bar{R}^{(k)}) \in \mathcal{F}_{MAR}$. Based on the acquired features at acquisition step $k$, a new observation action $\bar{U}^{(k)}$ is performed (here: $\bar{U}^{(k)} = $"observe($X_3$)").

In order to formulate the AFA problem, we employ an alternative view on missingness, the *sequence view*, which focuses on the sequence of feature acquisitions (Figure 1B). The sequence view arises from inducing a topological ordering in m-graphs, as follows. We assume that a missingness indicator $\bar{R}$ is formed during a sequence of observation actions $\bar{U} = (\bar{U}^{(1)}, ..., \bar{U}^{(N)})$, where action $\bar{U}^{(k)}$ can take on values in {"observe($X_1$)","observe($X_2$)",...} if $k < N$. The last observation action is always defined as $\bar{U}^{(N)} = $ "stop observation". Thus, $\bar{R}^{(k)}$ (resp. $\bar{X}^{(k)}$) corresponds to the

intermediate missingness indicator (resp. intermediate observed proxy) after $k$ steps of observation. We denote the distribution $p(\bar{U}^{(k)}|X, \bar{R}^{(k)}) \equiv \pi(\bar{U}^{(k)}|X, \bar{R}^{(k)})$ as the missingness policy. In order to introduce an AFA policy later, we need to define a restriction to the space of policies as follows. We define the functional space $\mathcal{F}_{MAR}$ (for MAR policies $\pi_{MAR} \in \mathcal{F}_{MAR}$) by restricting the missingness policy to be dependent only on data observed up to the current time step $k$: $\pi_{MAR}(\bar{U}^{(k)}|X, \bar{R}^{(k)}) = \pi_{MAR}(\bar{U}^{(k)}|\bar{X}^{(k)}, \bar{R}^{(k)})$. This means a MAR policy cannot base acquisition decisions on the values of not (yet) observed features. Note that this is a special case of MAR in which observations depend on the earlier variables in the m-graph under the sequence view.

### 3.2 Problem Formulation: Active feature acquisition (AFA)

The AFA problem without retrospective missingness is visualized as a causal graph in Figure 2. The goal of AFA is to find i) an optimal AFA agent that decides which features to acquire and ii) a prediction model which uses the acquired features to perform a prediction. We assume here a classification task as the subsequent prediction task, but the problem extends naturally to other tasks.

**Data** (Figure 2 bottom right):

Consider a $n_D \times (d_x + 1)$ dataset comprised of $n_D$ i.i.d. realizations of a $d_x$-dimensional random vector of features $X \in \mathbb{R}^{d_x}$ and a 1-dimensional random variable of the label $Y \in \mathbb{N}$. We consider $X$ and $Y$ fully observed for now and denote them as the environment of the AFA agent. The AFA graph shows an arrow $X \rightarrow Y$ (the features cause the label), but all results in this paper also hold under a reversed causal relationship ($X \leftarrow Y$).

**AFA missingness** (Figure 2 top left):
The AFA missingness is induced by the AFA agent that decides which subset of features to acquire. Let the missingness indicator $\hat{R}$ denote the AFA missingness in $X$ (which produces observed proxies $\hat{X}$). The corresponding policy $\pi(\hat{U}^{(k)}|X^{(k)}, \hat{R}^{(k)})$ is the AFA missingness policy. Our MAR assumption holds for the AFA missingness policy (i.e. $\pi(\hat{U}^{(k)}|X^{(k)}, \hat{R}^{(k)}) = \pi(\hat{U}^{(k)}|\hat{X}^{(k)}, \hat{R}^{(k)}) \in \mathcal{F}_{MAR}$). Further, consider a predefined, fixed acquisition cost $c_{acq}(i)$ of observing a feature $X_i$. The *acquisition cost* random variable $\hat{C}$ represents the cost that the AFA policy exerts when observing features in $X$: $\hat{C} = \sum_{i=1}^{d_x} \hat{R}_i \cdot c_{acq}(i)$. We make one important assumption about the application of the AFA agent at run-time/deployment:

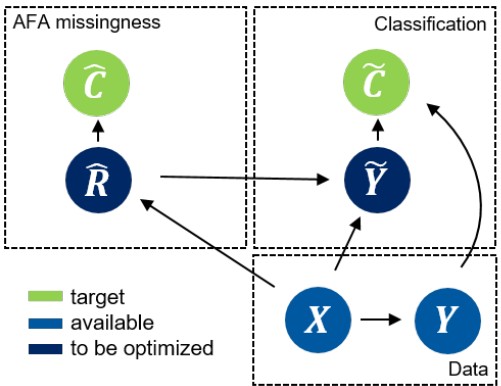

Figure 2: Causal graph for AFA: The AFA agent decides which of the features in $X$ to acquire (which produces $\hat{R}$) and pays feature acquisition costs $\hat{C}$. Based on the acquired set of features, the classifier predicts $\tilde{Y}$ and pays a misclassification cost $\tilde{C}$ if $Y$ and $\tilde{Y}$ mismatch.

**Assumption 1** *The AFA agent has full control of missingness at run-time.*

Assumption 1 implies that there are no other mechanisms that prevent the feature acquisition when the AFA agent requests the acquisition of a feature, i.e. the features are available for the agent, when asked for.

**Classification** (Figure 2 top right):
Consider a predicted label random variable $\tilde{Y} = f(\hat{R}, X) = f(\hat{R}, \hat{X})$ that is predicted based on the observed features $\hat{X}$ using classifier $f$. Further, consider wrong predictions produce a misclassification cost $\tilde{C} = c_{mc}(\tilde{Y}, Y)$, where $c_{mc}(y_l, y_m) : \mathbb{N}^2 \rightarrow \mathbb{R}$ is the cost of predicting a label of class $y_l$ as belonging to class $y_m$.

**AFA objective:**
The goal of AFA is to find a missingness policy $\pi_\theta(\hat{U}^{(k)}|\hat{X}^{(k)}, \hat{R}^{(k)})$, parameterized by $\theta$ and a classifier $f_\phi(\hat{R}, \hat{X})$, parameterized by $\phi$, that minimize the total sum of acquisition and

misclassification costs $C = \hat{C} + \tilde{C}$. The associated objective function is defined as

$$\arg\min_{\phi,\theta} J(\phi,\theta) = \arg\min_{\phi,\theta} \mathbb{E}\left[C\middle|\phi,\theta\right] \tag{1}$$

where $J(\phi,\theta)$ denotes the expected total cost after applying AFA missingness policy and classifier. Based on the described sequential nature of the AFA problem, one may find an optimal AFA agent using a method of choice, for example RL. In this work, however, we focus on the important issue of AFAPE, i.e. the unbiased evaluation of AFA agents (via estimating $J$) under retrospective missingness, which arises when the available dataset contains missingness. For ease of reference, we drop $\phi$ and $\theta$, the arguments of the $J$ notation.

### 3.3 Problem Formulation: Active Feature Acquisition Performance Evaluation (AFAPE) under Missing Data

The objective of evaluating AFA agents is to estimate the expected cost $J$ in Eq.(1). For a fully observed dataset, estimation is trivially done by sampling from the causal graph in Figure 2 (i.e. running both the agent and the classifier on the dataset) and averaging the resulting costs. Under retrospective missingness, however, one cannot sample from $p(X, Y)$, as some of the features in X are missing. We assume retrospective missingness scenarios without missingness in the label $Y$ and no direct influence of $Y$ on missingness (i.e. no arrow $Y \rightarrow \bar{R}$), but these restrictions can also be relaxed. Note that, in contrast to the AFA missingness (denoted by $\hat{R}$, $\hat{U}$ and $\hat{X}$), the retrospective missingness (denoted by $\bar{R}$, $\bar{U}$ and $\bar{X}$) is not necessarily restricted to $\mathcal{F}_{MAR}$.

A possible strategy to still run the agent under retrospective missingness is to allow the agent to acquire only the available features and block the acquisition action if the feature is not available (i.e. if $\bar{R}_i = 0$, then $\hat{R}_i = 0$). This results in a modified causal graph, the *AFA graph* (Figure 3), which comprises the retrospective missingness indicator $\bar{R}$ as an environment variable. In Appendix A.3 we provide details about sampling from the AFA graph. The arrow $\bar{R} \rightarrow \hat{R}$ restricts the AFA policy to observe only features that are available. The arrow $X \rightarrow \bar{R}$ corresponds to the retrospective missingness mechanism which is itself represented by an m-graph (as described in Figure 1A).

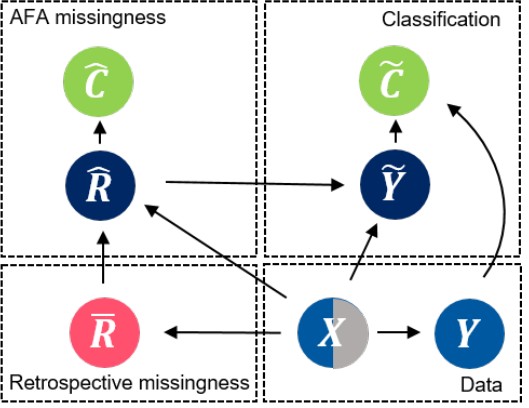

Figure 3: AFA graph: Causal graph for AFA setting with retrospective missingness. The distribution shift corresponds to an intervention $do(\bar{R} = \vec{1})$.

Assumption 1, which assumes full control of the AFA agent over the missingness at run-time, implies that the AFA agent is allowed to implement causal interventions on $\bar{R}$. This is contrary to the situation shown in Figure 3, where $\bar{R}$ depends on $X$. This causal intervention is mathematically expressed as $do(\bar{R} = \vec{1})$. As a result, we define the problem of active feature acquisition performance evaluation (AFAPE) under missing data as the task of evaluating

$$J_{do(\bar{R}=\vec{1})} = \mathbb{E}\left[C\middle|do(\bar{R} = \vec{1})\right]. \tag{2}$$

In the causal graph in Figure 3, $X$ is a confounder (there exists a path $\bar{R} \leftarrow X \rightarrow C$) which requires adjustment.

### 3.4 Solutions to AFAPE

We now examine different views and estimation strategies for the AFAPE problem that make use of the different views on missingness and the variables in the AFA graph.

### 3.4.1 BLACK-BOX MISSING DATA VIEW

We call the first view the black-box missing data view. It corresponds to the traditional view on missingness adjustment from the missing data literature, which hasn't been applied in AFA settings previously. It reduces the AFA graph to the three variables $X$, $\bar{R}$ and $C$, as shown in Figure 4A). We denote it as 'black-box' since it makes no use of the special structure of the AFA problem (knowledge about $\hat{R}$) and treats the AFA agent and classifier as a black-box function that maps $\bar{R}$ and $X$ to $C$.

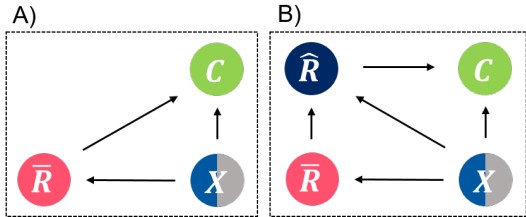

Figure 4: Reduced AFA graph for different estimation strategies. A) Black-box missing data view. B) Off-environment bandit view.

The black-box missing data view leads to the following IS-based estimator for $J_{do(\bar{R}=\vec{1})}$, called inverse probability weighting (IPW) (Seaman & White, 2013):

$$J_{IPW} = \mathbb{E}_{C,X,\bar{R}}\left[ C \frac{\mathbb{I}(\bar{R}=\vec{1})}{p(\bar{R}=\vec{1}|X)} \right] = \mathbb{E}_{C,X_{adj},\bar{R}}\left[ C \frac{\mathbb{I}(\bar{R}=\vec{1})}{p(\bar{R}=\vec{1}|X_{adj})} \right] \tag{3}$$

where $X_{adj}$ is a subset of features that blocks all backdoor paths between $X$ and $C$.

**Intuition behind IPW:**
The importance ratio $\rho_{IPW} = \frac{\mathbb{I}(\bar{R}=\vec{1})}{p(\bar{R}=\vec{1}|X)}$ (where $p(\bar{R}=\vec{1}|X)$ denotes the propensity score) is derived from the importance ratio $\frac{p(X|do(\bar{R}=\vec{1}))}{p(X|\bar{R}=\vec{1})}$. The idea of this IS approach is to sample from the known distribution $p(X|\bar{R}=\vec{1})$ instead of the unknown target feature distribution $p(X|do(\bar{R}=\vec{1}))$ and to adjust for the difference using the so-called importance ratio.

**Advantage of IPW:**
A big advantage of IPW is its completeness property, i.e. any function of the target distribution is identified if and only if the propensity score is identified (Nabi et al., 2020). Therefore, if identification for a particular missingness scenario fails for this approach, no other approach can perform unbiased estimation.

**Disadvantage of IPW:**
Under IPW, AFA agents can only be tested on the complete subset of the data (the IS distribution is $p(X|\bar{R}=\vec{1})$). This is a severe disadvantage, as even an agent that never acquires a certain feature could nevertheless be evaluated only on data points where that feature has observed entries.

We point out another common estimation strategy in the missing data literature: multiple imputation (MI) (Sterne et al., 2009) which we review in Appendix A.4. This approach can be adopted to overcome the data inefficiency of IPW. However, it suffers from the disadvantage of a potentially very complicated identification step and the requirement for fitting many conditional densities, which increases the risk of model misspecification. In Appendix A.4, we also discuss common mistakes such as imputing using only features (instead of features and label) and conditional mean imputation, which both can lead to biased results.

### 3.4.2 OFF-ENVIRONMENT BANDIT VIEW

We now propose a second view for AFAPE that leverages the knowledge about the AFA feature acquisitions $\hat{R}$ (Figure 4B). Although AFA was defined as a sequential decision problem, the order of acquisitions does not influence the evaluation step as the acquisition and misclassification costs are invariant to the order of acquisitions. We therefore treat the whole set of acquisitions $\hat{R}$ as one action and the agent as a multiarmed bandit (Auer et al., 2002). Thus, $\bar{R}$, $X$ and $Y$ can be seen as the environment, while $\hat{R}$ denotes the bandit's action and $C$ the return. We thus take the set view for both the retrospective and the AFA missingness. The AFAPE problem can thus be seen as off-environment policy evaluation of a bandit under an environment intervention.

Based on the off-environment bandit view, we propose AFAIS (active feature acquisition importance sampling), a novel unbiased and consistent estimator for $J_{do(\bar{R}=\vec{1})}$:

$$J_{AFAIS} = \mathbb{E}_{C,\hat{R},X_{adj}} \left[ C \frac{p(\bar{R}=\vec{1}|X_{adj},\hat{R})}{p(\bar{R}=\vec{1}|X_{adj})} \right] \tag{4}$$

The derivation is shown in Appendix A.5.

**Intuition behind AFAIS:**
The term $\rho_{AFAIS} = \frac{p(\bar{R}=\vec{1}|X_{adj},\hat{R})}{p(\bar{R}=\vec{1}|X_{adj})}$ is derived from the importance ratio $\frac{p(\hat{R}|X_{adj},\bar{R}=\vec{1})}{p(\hat{R}|X_{adj})}$. The idea behind this approach is that we switch the adjustment from different feature distributions (as in IPW) to an adjustment for different AFA feature acquisition distributions. The problem remains that we cannot sample from the policy $p(\hat{R}|X_{adj}, \bar{R} = \vec{1})$ for every data point because we do not have certain features (and have to block those actions). Instead, we sample from $p(\hat{R}|X_{adj}) = \sum_{\bar{R},X_{-adj}} p(\hat{R}|X_{adj}, X_{-adj}, \bar{R}) p(\bar{R}, X_{-adj}|X_{adj})$ which is possible to do over all data points (not just fully observed data points). The variable set $X_{-adj}$ corresponds to the subset of features not in $X_{adj}$.

**Advantage of AFAIS:**
The obvious and main advantage of AFAIS is that it uses all data points instead of only complete cases (as in IPW) which increases data efficiency. However, this advantage diminishes in case of "data hungry" agents, i.e. agents that often acquire relatively large feature sets. For instance, if the AFA agent always acquires all the features, i.e. $p(\hat{R} = \vec{1}|X_{adj}, \bar{R} = \vec{1}) = 1 \ \forall X_{adj}$, then the numerator, which we denote as the AFA propensity score, $p(\bar{R} = \vec{1}|X_{adj}, \hat{R}) = 0$ if $\hat{R} \neq \vec{1}$ and AFAIS and IPW will have equal weights.

**Disadvantage of AFAIS:**
One disadvantage of AFAIS is that it requires the fitting of a second function, the AFA propensity score $p(\bar{R} = \vec{1}|X_{adj}, \hat{R})$, which increases the chance of model misspecification. Additionally, identification is more demanding for AFAIS. In Appendix A.6, we show the identifiability of $J_{AFAIS}$ under MCAR and simple MAR missingness scenarios, and show how it cannot be evaluated in an MNAR setting.

### 3.4.3 THE AFAIS-IPW SPECTRUM

AFAIS and IPW can be viewed as two ends of a spectrum. In IPW, we only use the complete cases, while in AFAIS we use all data points with all missingness patterns. A tuning parameter that allows us to gradually move on this spectrum toward each approach, helps us overcome the disadvantages of both (low data efficiency for IPW and problem of evaluation under MNAR missingness for AFAIS). We propose the modified AFAIS estimator as:

$$J_{AFAIS}(s) = \mathbb{E}_{C,\hat{R},X_{adj},\bar{R}_s} \left[ C \frac{p(\bar{R}=\vec{1}|X_{adj},\hat{R},\bar{R}_s=\vec{1})\mathbb{I}(\bar{R}_s=\vec{1})}{p(\bar{R}=\vec{1}|X_{adj})} \right] \tag{5}$$

for any desired index subset $s \subseteq \{0, \ldots, d_x\}$. The derivation is shown in Appendix A.7.

**Intuition behind the modified AFAIS:**
The idea behind this approach is to sample only from data points where an important set of features (the features in $s$) are always observed while using the off-environment bandit approach for the remaining features. To showcase the advantages of using the modified AFAIS, we demonstrate in Appendix A.8 that the modified AFAIS is identified for the MNAR example from Appendix A.6 and that it can be evaluated when an appropriate conditioning set $s$ is chosen. We do not give general identifiability statements for the modified AFAIS estimator, but suggest a simple procedure for choosing the set $s$ in Appendix A.8.

### 3.4.4 OTHER VIEWS ON SOLVING AFAPE AND DATA EFFICIENCY

Other views on AFAPE include off-policy policy evaluation (OPE) which has been used in the AFA literature before, but is very data inefficient in the context of AFA and can lead to biased results in complex missing data scenarios. Another approach used in the AFA literature is to just block

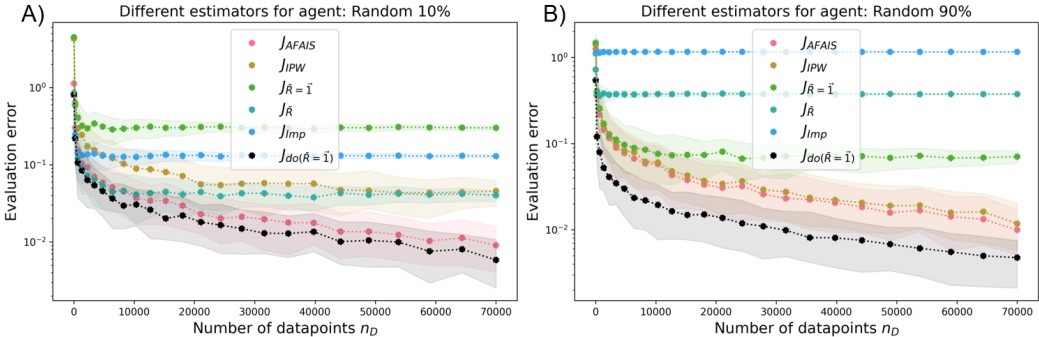

Figure 5: Error of expected cost estimation as a function of the number of data points in a synthetic data experiments with MNAR missingness. Plots show two random agents that acquire each costly feature with a probability of A) 10% and B) 90%. Error is defined as the difference between an estimation and the ground truth estimator $J_{do(\bar{R}=\vec{1})}$ evaluated on the whole dataset ($n_D = 100000$).

the acquisitions without adjustment which thus also leads to biased results. We review these two approaches in Appendix A.9. In Appendix A.10 we quantify the improved data efficiency of the off-environment bandit view over the black-box missing data and OPE views. We also show that the improved data efficiency leads to weaker positivity assumptions required for the off-environment bandit view compared to the other views.

# 4 EXPERIMENTS

## 4.1 EXPERIMENT DESIGN

We evaluate different estimators on synthetic and real world datasets (HELOC (FICO, 2018), Retinopathy (Antal & Hajdu, 2014), and Income (Newman et al., 1998)) under examples of synthetic MCAR, MAR and MNAR retrospective missingness. We evaluate random AFA policies and a vanilla Deep Q-Network (DQN) RL agent (Mnih et al., 2015) as AFA agents and use a random forest classifier. We compare the following estimators:

- $J_{Imp}$ uses Missforest (Stekhoven & Bühlmann, 2012) to impute missing features.
- $J_{\bar{R}}$ blocks the acquisitions of not available features, but offers no correction.
- $J_{\bar{R}=\vec{1}}$ corresponds to complete case analysis and is only unbiased under MCAR.
- $J_{IPW}$ is the stabilized IPW estimator. Stabilization is achieved by normalizing the weights which reduces variance at the cost of small bias (Rubinstein & Kroese, 2016).
- $J_{AFAIS}$ is the stabilized (modified) AFAIS estimator.
- $J_{do(\bar{R}=\vec{1})}$ is the ground truth, where the agent is run on the fully observed dataset.

We fit the propensity score and AFA propensity score using multi-layer perceptrons. Complete experiment details are given in Appendix A.11.

## 4.2 RESULTS

Figure 5 depicts convergence rates of different estimators for the MNAR synthetic data experiment for two random policies as a function of evaluation sample size. The results show that the three knowingly-biased estimators $J_{Imp}$, $J_{\bar{R}}$ and $J_{\bar{R}=\vec{1}}$ do not converge to correct solutions as the sample size increases. The stabilized AFAIS (and IPW) estimators converge to the correct solution, but at different rates depending on the policy. In the case of a policy that only acquires 10% of the costly features, the stabilized AFAIS estimator $J_{AFAIS}$ is converging almost as fast as the ground truth $J_{do(\bar{R}=\vec{1})}$ (Figure 5 A). In the case where the agent acquires 90% of the data, the AFAIS and IPW estimators are converging at similar rate, indicating there is not much benefit of using

AFAIS over the conventional IPW estimator (Figure 5 B). The increased data efficiency of AFAIS is illustrated in Figure 6 on real-world data experiments where it leads to tighter tighter boxplots for AFAIS compared to IPW. This benefit is significant for agents that do not acquire much data (i.e. the random $10\%$ agent), and diminishes as the agent becomes more "data hungry". The experiments results in Figure 6 further demonstrate the large errors biased estimators, such as $J_{Imp}$, $J_{\bar{R}}$ and $J_{\bar{R}=\vec{1}}$, can produce and the potentially wrong conclusions these estimators can lead to. If the policies are compared according to $J_{Imp}$, for example, one would conclude the differences between the random policies in the HELOC dataset are large ($J_{Imp}$ ranges between 6.5 and 8), when in fact they are only small ($J_{do(\bar{R}=\vec{1})}$ is close to 6.5 for all random policies). Further experiment results are shown in Appendix A.12.

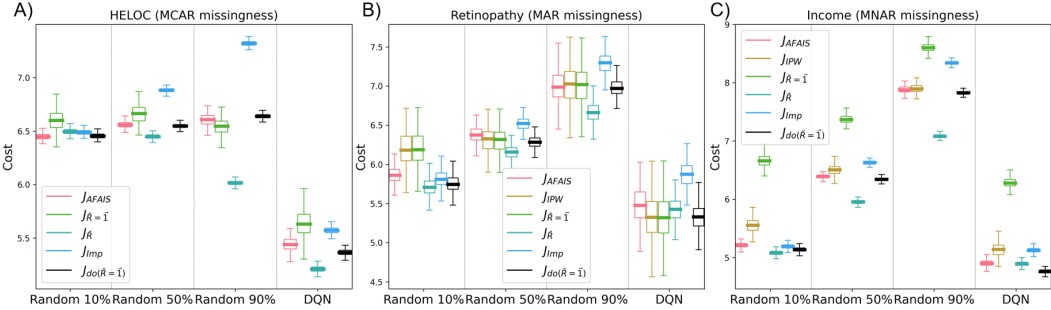

Figure 6: Results for the A) HELOC, B) Retinopathy, and C) Income datasets for different missingness scenarios. Boxplots show variability of the estimate when bootstrapping 300 times without replacement 70% of a total test dataset of size $n_{D_{test}}$ ($n_{D_{test}} = 17'781$ for HELOC, $n_{D_{test}} = 575$ for Retinopathy and $n_{D_{test}} = 16'281$ for Income).

## 5    DISCUSSION AND FUTURE WORK

The central argument of this paper is based on Assumption 1, the full control of the AFA agent over the missingness at run-time. This assumption might, however, not always hold in practice (e.g. due to the breaking of a sensor or a patient who is not cooperative). In this case, a different distribution shift for $\bar{R}$ should be assumed for evaluation.

In general, the choice of view and estimator for AFAPE must depend on the following questions:
1) *Can identification be performed?* The identification step might have varying degrees of difficulty depending on the underlying m-graph and the chosen view/ estimator. Identification is for example easier for IPW than for AFAIS.
2) *Which estimator is most data efficient?/ Do all positivity assumptions hold?* In this paper, we demonstrate that the off-environment bandit view is more data efficient than the black-box missing data and OPE views. Furthermore, it requires the weakest positivity assumptions.
3) *Can the nuisance functions be fitted accurately?* The nuisance functions (e.g. the AFA propensity score) need to be fitted based on the available data. Fitting the AFA propensity score can thus pose as an additional source of error of the AFAIS estimator that the IPW estimator does not have.

We consider the following suggestions as major extensions of this work: i) studying the evaluation of AFA agents under missingness in the temporal setting with time-series features; ii) including the counterfactual expected cost $J_{do(\bar{R}=\vec{1})}$ in the training of AFA agents and classifiers.

## 6    CONCLUSION

We study the problem of active feature acquisition performance evaluation (AFAPE) under missing data which corresponds to an intervention on the environment used to train AFA agents. We propose a new causal graph, the AFA graph, and demonstrate that current evaluation methods for the AFAPE problem can be strongly biased leading to high risks for model deployment. We show that AFAIS, a novel IS-based estimator is an unbiased estimator for evaluation, and more data efficient than IPW.

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

## A  APPENDIX

### A.1  LITERATURE REVIEW FOR ACTIVE FEATURE ACQUISITION

Active feature acquisition (AFA) (An et al., 2006; Li & Oliva, 2021a; Li et al., 2021; Chang et al., 2019; Shim et al., 2018; Yin et al., 2020) has been studied under various other names including, but not limited to, "active sensing" (Yoon et al., 2019; 2018; Tang et al., 2020; Jarrett & van der Schaar, 2020), "active feature elicitation" (Natarajan et al., 2018; Das et al., 2021),"dynamic feature acquisition" (Li & Oliva, 2021b), "dynamic active feature selection" (Zhang, 2019), "element-wise efficient information acquisition" (Gong et al., 2019), "classification with costly features" (Janisch et al., 2020) and "test-cost sensitive classification" (Xiaoyong Chai et al., 2004). AFA is different from active learning (Settles, 2009). In active learning, one assumes a classification task with a training data set that contains many unlabeled data points. The active learning task is then to decide which label acquisitions will improve the training performance the most. Similar research also exists for the acquisition of features for optimal improvement of training. This task has been referred to as "active selection of classification features" (Kok et al., 2021), and unfortunately also as "active feature acquisition" (Huang et al., 2018; Beyer et al., 2020), but its objective differs fundamentally from ours. Huang et al. (2018) attempt to find out which missing values within the retrospective data set would improve training the most when retroactively acquired. In this paper, we are, however, interested which features, for a new data point, would improve the individual prediction for that data point the most.

Many AFA approaches are based on a greedy feature acquisition strategy wrapping a subsequent classification task. An idea is to employ decision tree classifiers and to acquire features sequentially by traversing the branch of the decision tree (Ling et al., 2004; Sheng & Ling, 2006), while the splitting criteria minimizes the combined cost of feature acquisition and misclassification (Ling et al., 2004). Another approach, the test-cost sensitive Naive Bayes (csNB) classifier (Xiaoyong Chai et al., 2004), exploits the Naive Bayes assumption of independence among the predictive power of features. This allows for an efficient exploration whether acquiring a certain feature can reduce

costs. Das et al. (2021) propose clustering-based cost-aware feature elicitation (CATE). In CATE, data points are clustered based on a set of zero-cost features and the optimal, fixed set of features is computed for each cluster. A new partially-observed data point is then attributed to a cluster, and the corresponding optimal feature set is acquired for it.

The AFA problem is inherently a sequential decision process which one can address using reinforcement learning (RL). Model-based RL approaches learn the state-transition function which, for AFA, corresponds to learning an imputation model (Yoon et al., 2018; Yin et al., 2020; Li & Oliva, 2021a;b; Ma et al., 2019). The imputation model is used at run-time to simulate possible outcomes of a feature acquisition and derive desired acquisition strategies. Alternatively, model-free RL approaches do not require learning a state-transition function. One variant, Q-learning, relies on modeling the expected cost of a particular acquisition decision (Chang et al., 2019; Janisch et al., 2020; Shim et al., 2018). As an example, Shim et al. (2018) use double Q-learning for the AFA agent with a deep neural network that shares network layers with the subsequent classification neural network.

## A.2 MISSING DATA MODELS

Missing data problems relate the full data distribution $p(X)$ and the observed data distribution where variables in $X$ may sometimes be unobserved. Each $X_i \in X$ is observed if the corresponding missingness indicator $\bar{R}_i$ assumes value 1. Thus, the observed version of $X_i$ which we denote by $\bar{X}_i$ is defined to be equal to $X_i$ if $\bar{R}_i = 1$, and is missing (equal to a special value "?") otherwise. We further consider an acquisition cost $\bar{C}_i$ for the feature $X_i$. The cost $\bar{C}$ is thus deterministically defined by the missingness indicator $\bar{R}$. The goal in missing data problems is to estimate some function of $p(X)$, such as a parameter or a predictive model, using the i.i.d. samples from the observed distribution $p(\bar{X}, \bar{R})$, where $\bar{X}$ is the set of all observed proxies, and $\bar{R}$ is the set of all missingness indicators in the problem. Parameters of interest must be identified, i.e. unique functions of $p(\bar{X}, \bar{R})$, in order to make the estimation problem well-posed.

The problem of identification is to determine whether or not a function of the full data can be recovered from the observed data distribution. Identification is not possible in general, and relies on assumptions encoded in a missing data model. The simplest identified missing data model is the missing completely at random (MCAR) model, which assumes that $p(\bar{R}|X) = p(\bar{R})$. The missing at random model (MAR) assumes that missingness only depends on observed features: $p(\bar{R} = \bar{r}|X) = p(\bar{R} = \bar{r}|\{X_i : \bar{r}_i = 1\})$. Finally, any model that is neither MCAR nor MAR is called missing not at random (MNAR) where missingness can now depend also on feature values that are unobserved. While parameters of interest are not identified in general under MNAR, a large class of identified MNAR models has been derived in the missing data literature (Bhattacharya et al., 2020; Nabi et al., 2020).

In this paper, we will restrict attention to a special type of missing data models where identifiability restrictions are represented by a directed acyclic graph (DAG) factorization of the distribution. The restrictions are then formalized as $p(X, \bar{R}) = \prod_{V \in X \cup \bar{R}} p(V|\text{pa}_{\mathcal{G}}(V))$ for some graph $\mathcal{G}$ where $\text{pa}_{\mathcal{G}}(V)$ selects the parents of the node $V$ in $\mathcal{G}$. The graph $\mathcal{G}$ is termed the *missing data graph* (m-graph) (Mohan et al., 2013; Shpitser et al., 2015) (Figure 1A). In models of missing data, the identification problem for parameters in $p(X)$ from $p(\bar{X}, \bar{R})$ may be formulated as an identification problem of causal inference, where $p(X)$ is obtained as a result of applying the intervention operator $\text{do}(.)$ (Pearl, 2009) to every missingness indicator $\bar{R}$, hence $p(X) = p(\bar{X}|\text{do}(\bar{R} = \vec{1})) = \frac{p(\bar{X}, \bar{R}=\vec{1})}{p(\bar{R}=\vec{1}|X)}$. In other words, the problem of identification of $p(X)$ is translated to the problem of identification of $p(\bar{R} = \vec{1}|X)$. If the missing data model is graphical, techniques from causal effect identification theory may be applied to obtain identification of $p(X)$ (Bhattacharya et al., 2020; Nabi et al., 2020).

## A.3 SAMPLING FROM THE AFA GRAPH

In this section, we provide the details of factorizing the joint distribution over the variables of the AFA process with respect to the AFA graph (Figure 3) into distributions from which we can sample. The joint distribution factorizes as

$$p(\tilde{C}, \tilde{Y}, \hat{C}, \hat{R}, \bar{R}, X, Y) = p(\tilde{C}|\tilde{Y}, Y)p(\hat{C}|\hat{R})p(\tilde{Y}|X, \hat{R})p(\hat{R}|X, \bar{R})p(\bar{R}, X, Y). \quad (\text{A.1})$$

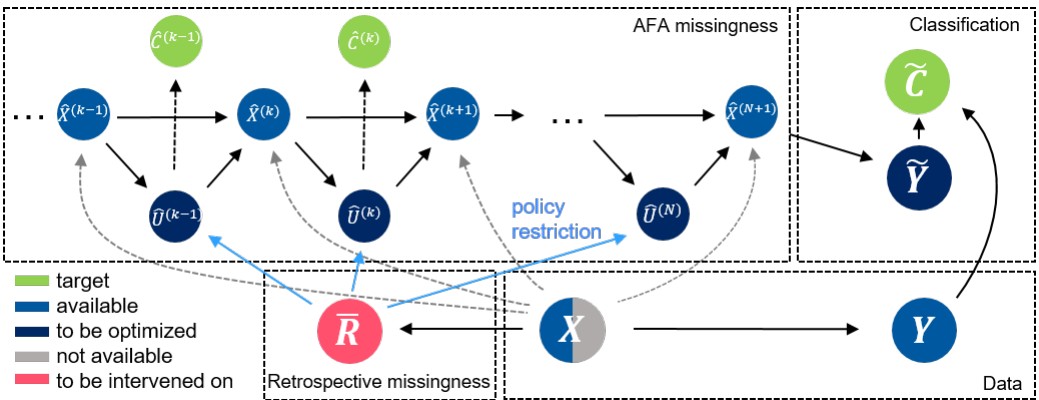

Figure A.1: AFA graph from an AFA missingness policy viewpoint. The node $\hat{X}^{(k)}$ represents the intermediate states at step $k$ of the decision process. The state also contains $\hat{R}^{(k)}$ which was left out for clarity. The AFA policy $\pi(\bar{U}^{(k)}|\hat{X}^{(k)}, \hat{R}^{(k)}, \bar{R})$ is restricted by $\bar{R}$ to only allow the acquisition of available features, i.e. $\pi(\bar{U}^{(k)} = \text{"observe}(X_i)\text{"}|\hat{X}^{(k)}, \hat{R}^{(k)}, \bar{R} = \bar{r}) = 0$ if $\bar{r}_i = 0$.

The sampling procedure thus is defined as follows. First, we sample from the retrospective dataset $p(\bar{R}, X, Y)$ (i.e. the "environment"). Next, we run the AFA agent. This gives us samples from the AFA missingness $p(\hat{R}|X, \bar{R})$ and the (deterministic) acquisition cost $p(\hat{C}|\hat{R})$ conditional distributions. As AFA missingness is not modelled from a set viewpoint (i.e. we cannot directly sample $p(\hat{R}|X, \bar{R})$), we sample equivalently from the corresponding AFA policy $\pi(\hat{U}^{(k)}|\hat{X}^{(k)}, \hat{R}^{(k)}, \bar{R})$. To visualize this process, we present in Figure A.1 the AFA graph from an AFA missingness sequence perspective. Sampling $\hat{R}$ now corresponds to sampling $\hat{U}$. We block feature acquisition actions if the desired feature is not available (i.e. $p(\hat{R} = \hat{r}|X, \bar{R} = \bar{r}) = 0$ if $\hat{r}_i = 1$ and $\bar{r}_i = 0$). Finally, we apply the (deterministic) classifier and compute the misclassification cost (i.e. "sample" from $p(\tilde{Y}|X, \hat{R})$ and $p(\tilde{C}|\tilde{Y}, Y)$).

### A.4 MULTIPLE IMPUTATION (MI) FOR THE AFAPE PROBLEM

An alternative estimation strategy to IPW based on the missing data view is multiple imputation (MI). The core idea of imputation is to fill in (impute) missing entries and then assume a fully observed dataset. The procedure can be formulated as (Little & Rubin, 2019)

$$J_{do(\bar{R}=\vec{1})} = \mathbb{E}_{X,Y}[h(X, Y)] = \mathbb{E}_{\bar{X},Y}\big[\mathbb{E}_{X_{miss}}[h(\bar{X}, X_{miss}, Y)]\big] \qquad \text{(A.2)}$$
$$= \sum_{\bar{X},Y} \sum_{X_{miss}} h(\bar{X}, X_{miss}, Y)p(X_{miss}|\bar{X}, Y)p(\bar{X}, Y)$$

where $h(X, Y) \equiv C$ is the black-box function mapping the data to the costs and $X_{miss}$ represents the unobserved data. Imputation thus corresponds to Monte Carlo integration by sampling observed values $p(\bar{X}, Y)$ and imputing missing values using an imputation model $p(X_{miss}|\bar{X}, Y)$. To increase accuracy of the estimator, the imputation can be repeated multiple times leading to the MI procedure. In the identification step, one determines whether the imputation model can be estimated from retrospective data.

The first drawback of imputation is the requirement of modeling joint distributions which is in practice a complex problem, especially in high-dimensional settings and under complex missingness patterns. As an example, the multiple imputation by chained equations (MICE) method (van Buuren, 2007) requires fitting $K$ conditional densities for $K$ partially observed features. By comparison, IPW requires only the propensity score, which is often much easier to specify.

Eq.(A.2) implies imputation by conditioning on the features $X$ and the label $Y$. This introduces the risk of data leakage, as the imputed features may carry predictive information not because of the true data generation mechanism, but because of the imputation, resulting in a potentially overoptimistic estimation of prediction performance. A common alternative, often used in ML, is to impute the data

without conditioning on $Y$. This assumption, however, implies that a missing feature $X_i \in X_{miss}$ is conditionally independent of the label given the observed features ($X_i \perp\!\!\!\perp Y|\bar{X}$). Determining marginal predictive value of a feature for predicting $Y$, is however, the whole task of AFA, which renders this approach impractical.

A simplified imputation approach that reduces the complexity of modelling and that has been applied in AFA settings (An et al., 2022; Erion et al., 2021; Janisch et al., 2020) is conditional mean imputation. The missing values are thereby imputed using a conditional mean model for $\mathbb{E}[X_{miss}|\bar{X}]$. Conditional mean imputation assumes $\mathbb{E}_{\bar{X},Y}[\mathbb{E}_{X_{miss}}[h(\bar{X}, X_{miss}, Y)]] = \mathbb{E}_{\bar{X},Y}[h(\bar{X}, \mathbb{E}_{X_{miss}}[X_{miss}|\bar{X}], Y)]$, which does not hold in general and can lead to strongly biased results when $h$ is nonlinear as is the case in AFA settings.

## A.5 DERIVATION OF ACTIVE FEATURE ACQUISITION IMPORTANCE SAMPLING (AFAIS)

The AFAIS estimator is based on IS, a popular method in OPE and missing data to adjust for distribution shift. IS assumes a problem setting where one has only access to samples from a "behavior" distribution, while the goal is the evaluation of a "target" distribution. This is made possible when knowledge of the ratio of the two distributions (importance ratio) is available. In the setting of this paper, the target distribution is the distribution of $C$ under the intervention $do(\bar{R} = \vec{1})$. The derivation of AFAIS is based on the reduced AFA graph from Figure 4B) which results from a latent projection (Verma & Pearl, 1990) of the full AFA graph from Figure 3. This reduces the graph factorization from equation Eq. (A.1) to only the features $C$, $\hat{R}$, $\bar{R}$ and $X$. We further consider only a subset of the features $X_{adj}$ that blocks all backdoor paths from $\bar{R}$ to $\hat{R}$ and $C$. Since this yields a causal model of a DAG, we can apply the truncated factorization, or the g-formula, to obtain:

$$p(C, \hat{R}, X_{adj}|do(\bar{R} = \vec{1})) = p(C|X_{adj}, \hat{R})p(\hat{R}|X_{adj}, \bar{R} = \vec{1})p(X_{adj}) \tag{A.3}$$

We propose the following behavior distribution:

$$p(C, \hat{R}, X_{adj}) = p(C|X_{adj}, \hat{R})p(\hat{R}|X_{adj})p(X_{adj}) \tag{A.4}$$

This means we sample from all data points and not only from observed data points as for IPW. Sampling from Eq. (A.4) is equivalent to sampling from $p(C, \hat{R}, X_{adj}, X_{-adj}, \bar{R}) = p(C, \hat{R}, X, \bar{R}) = p(C|X, \hat{R})p(\hat{R}|X, \bar{R})p(X, \bar{R})$ which one can do as shown in Appendix A.3.

Based on the introduced target and behavior distribution, we now derive the AFAIS estimator:

$$J_{do(\bar{R}=\vec{1})} = \mathbb{E}\left[C\Big|do(\bar{R} = \vec{1})\right] = \sum_C Cp(C|do(\bar{R} = \vec{1}))$$

$$= \sum_{C,X_{adj},\hat{R}} Cp(C, X_{adj}, \hat{R}|do(\bar{R} = \vec{1}))$$

$$= \sum_{C,X_{adj},\hat{R}} Cp(C|X_{adj}, \hat{R})p(\hat{R}|X_{adj}, \bar{R} = \vec{1})p(X_{adj})$$

$$= \sum_{C,X_{adj},\hat{R}} Cp(C|X_{adj}, \hat{R})p(\hat{R}|X_{adj})p(X_{adj}) \cdot \frac{p(\hat{R}|X_{adj}, \bar{R} = \vec{1})}{p(\hat{R}|X_{adj})}$$

$$\stackrel{(*)}{=} \sum_{C,X_{adj},\hat{R}} Cp(C|X_{adj}, \hat{R})p(\hat{R}|X_{adj})p(X_{adj}) \cdot \frac{p(\bar{R} = \vec{1}|X_{adj}, \hat{R})}{p(\bar{R} = \vec{1}|X_{adj})}$$

$$= \mathbb{E}_{C,\hat{R},X_{adj}}\left[C\frac{p(\bar{R} = \vec{1}|X_{adj}, \hat{R})}{p(\bar{R} = \vec{1}|X_{adj})}\right] \equiv J_{AFAIS} \tag{A.5}$$

where the equivalence $(*)$ holds due to Bayes rule:

$$\frac{p(\hat{R}|X_{adj}, \bar{R} = \vec{1})}{p(\hat{R}|X_{adj})} = p(\hat{R}|X_{adj}, \bar{R} = \vec{1})\frac{p(\bar{R} = \vec{1}|X_{adj}, \hat{R})}{p(\hat{R}|X_{adj}, \bar{R} = \vec{1})p(\bar{R} = \vec{1}|X_{adj})} = \frac{p(\bar{R} = \vec{1}|X_{adj}, \hat{R})}{p(\bar{R} = \vec{1}|X_{adj})}$$

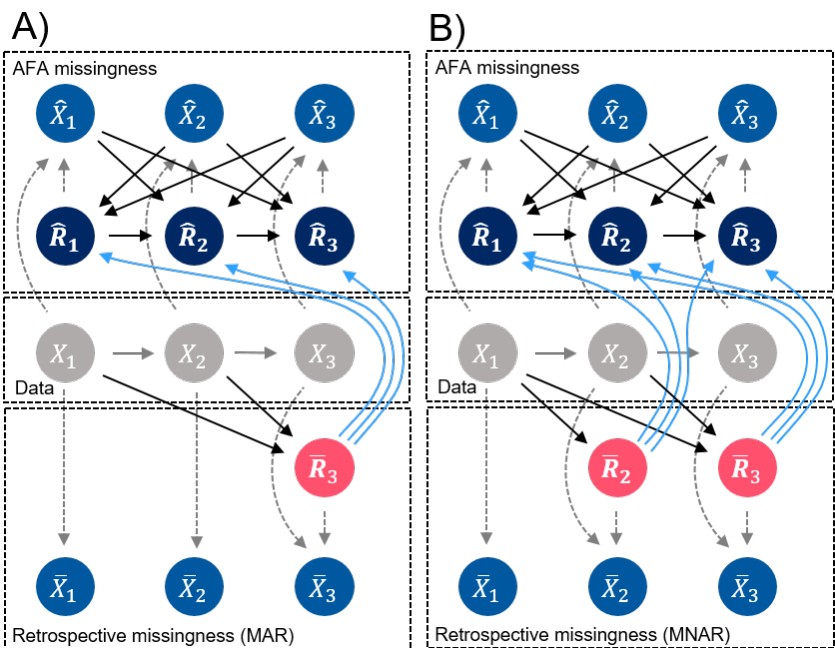

Figure A.2: Missing data graph including the AFA and retrospective missingness. A) Retrospective missingness is MAR. B) Retrospective missingness is MNAR.

### A.6 IDENTIFICATION EXAMPLE FOR AFAIS

We give an instructive example of how to perform identification for the propensity score $p(\bar{R} = \vec{1}|X_{adj})$ and AFA propensity score $p(\bar{R} = \vec{1}|X_{adj}, \hat{R})$, needed for the AFAIS estimator $J_{AFAIS}$. We show that identification is possible for a simple MAR (and therefore MCAR) scenario, but that the evaluation of the AFAIS estimator is not possible for an MNAR missing data scenario.

MAR

We assume the missing data graph in Figure A.2A) showing both retrospective and AFA missingness. The figure shows a simple MAR missing data scenario, where $X_1$ and $X_2$ are not missing in the retrospective dataset (we leave out $\bar{R}_1$ and $\bar{R}_2$ for clarity), but influence retrospective missingness in $X_3$ ($X_1 \rightarrow \bar{R}_3$; $X_2 \rightarrow \bar{R}_3$). The blocking of actions leads to: $\bar{R}_3 \rightarrow \hat{R}_i \, \forall i$. Furthermore, the AFA agent's decision to acquire a feature can be made based on all previously acquired features which translates to $\hat{X}_i \rightarrow \hat{R}_j \, \forall i \neq j$. Figure A.2 shows arrows $\hat{R}_1 \rightarrow \hat{R}_2$ and $\hat{R}_2 \rightarrow \hat{R}_3$, but this does not hold in general. Other causes between missingness indicators are in general possible, but this does not affect the identification procedure described in the following. The corresponding adjustment set is $X_{adj} = \{X_1, X_2\}$.

**Identification of $p(\bar{R} = \vec{1}|X_{adj})$:**
The propensity score is identified by:

$$p(\bar{R} = \vec{1}|X_{adj}) = p(\bar{R}_3 = 1|X_1, X_2)$$

which is a function of the observed data ($X_1$ and $X_2$ are always observed).

**Identification of $p(\bar{R} = \vec{1}|X_{adj}, \hat{R})$:**
The AFA propensity score is also identified:

$$p(\bar{R} = \vec{1}|X_{adj}, \hat{R}) = p(\bar{R}_3 = 1|X_1, X_2, \hat{R})$$

MNAR

We now assume the missing data graph in Figure A.2B). The missingness scenario is equal to the previously described MAR case, except for an additional missingness $\bar{R}_2$ caused by $X_1$ ($X_1 \rightarrow \bar{R}_2$). As the missingness indicator $\bar{R}_3$ now depends on a potentially missing $X_2$, the scenario is MNAR. The corresponding adjustment set remains $X_{adj} = \{X_1, X_2\}$.

**Identification of $p(\bar{R} = \vec{1}|X_{adj})$:**
The propensity score is still identified by

$$
\begin{aligned}
p(\bar{R} = \vec{1}|X_{adj}) &= p(\bar{R} = \vec{1}|X_1, X_2) \\
&= p(\bar{R}_2 = 1|X_1)p(\bar{R}_3 = 1|X_1, X_2) \\
&= p(\bar{R}_2 = 1|X_1)p(\bar{R}_3 = 1|X_1, X_2, \bar{R}_2 = 1) \quad\quad\text{(A.6)}
\end{aligned}
$$

where we used the conditional independence $\bar{R}_3 \perp\!\!\!\perp \bar{R}_2|X_1, X_2$ to make the expression a function of the observed data. The problem with using the propensity score for the AFAIS estimator is that it needs to be evaluated for all $X$ and not only for fully observed samples of $X$ (as is the case in regular IPW). In this example, it is not possible to evaluate $p(\bar{R} = \vec{1}|X_{adj})$ for data points where $\bar{R}_2 = 0$. This means AFAIS cannot be used. One could impute the missing values using the MI procedure. This can, however, be very difficult due to complex identification and modelling. Furthermore, the identification of the AFA propensity score might also be very difficult or potentially impossible in these scenarios.

### A.7 DERIVATION OF THE MODIFIED ACTIVE FEATURE ACQUISITION IMPORTANCE SAMPLING (AFAIS)

The derivation of the modified AFAIS estimator that combines importance sampling ideas from the black-box missing data view and the off-environment bandit view, and therefore covers the AFAIS-IPW spectrum, closely follows the derivation done in Appendix A.5. Again, the target distribution is

$$
p(C, \hat{R}, X_{adj}|do(\bar{R} = \vec{1})) = p(C|X_{adj}, \hat{R})p(\hat{R}|X_{adj}, \bar{R} = \vec{1})p(X_{adj}) \quad\quad\text{(A.7)}
$$

Now, we propose the following behavior distribution:

$$
p(C, \hat{R}, X_{adj}|\bar{R}_s = \vec{1}) = p(C|X_{adj}, \hat{R})p(\hat{R}|X_{adj}, \bar{R}_s = \vec{1})p(X_{adj}|\bar{R}_s = \vec{1}) \quad\quad\text{(A.8)}
$$

This means we sample from data points where some subset $s$ of the features is fully observed ($\bar{R}_s = 1$). Sampling from Eq. (A.8) is equivalent to samping from $p(C, \hat{R}, X_{adj}, X_{-adj}, \bar{R}) = p(C, \hat{R}, X, \bar{R}) = p(C|X, \hat{R})p(\hat{R}|X, \bar{R})p(X, \bar{R}|\bar{R}_s = \vec{1})$ which one can do as shown in Appendix A.3.

Based on the introduced target and behavior distribution, we now derive the modified AFAIS estimator:

$$
J_{do(\bar{R}=\vec{1})} = \mathbb{E}\left[C\Big|do(\bar{R} = \vec{1})\right] = \sum_C Cp(C|do(\bar{R} = \vec{1}))
$$

$$
= \sum_{C, X_{adj}, \hat{R}} Cp(C|X_{adj}, \hat{R})p(\hat{R}|X_{adj}, \bar{R} = \vec{1})p(X_{adj})
$$

$$
= \sum_{C, X_{adj}, \hat{R}} Cp(C|X_{adj}, \hat{R})p(\hat{R}|X_{adj}, \bar{R}_s = \vec{1})p(X_{adj}|\bar{R}_s = \vec{1}) \cdot \frac{p(\hat{R}|X_{adj}, \bar{R} = \vec{1})}{p(\hat{R}|X_{adj}, \bar{R}_s = \vec{1})}\frac{p(X_{adj})}{p(X_{adj}|\bar{R}_s = \vec{1})}
$$

$$
\overset{(*)}{=} \sum_{C, X_{adj}, \hat{R}} Cp(C|X_{adj}, \hat{R})p(\hat{R}|X_{adj}, \bar{R}_s = \vec{1})p(X_{adj}|\bar{R}_s = \vec{1}) \cdot \frac{p(\bar{R} = \vec{1}|X_{adj}, \hat{R}, \bar{R}_s = \vec{1})p(\bar{R}_s = \vec{1})}{p(\bar{R} = \vec{1}|X_{adj})}
$$

$$
= \mathbb{E}_{C, \hat{R}, X_{adj}, \bar{R}_s}\left[C\frac{p(\bar{R} = \vec{1}|X_{adj}, \hat{R}, \bar{R}_s = \vec{1})\mathbb{I}(\bar{R}_s = \vec{1})}{p(\bar{R} = \vec{1}|X_{adj})}\right] \equiv J_{AFAIS}(s) \quad\quad\text{(A.9)}
$$

where the equivalence $(*)$ holds due to Bayes rule:
The first fraction is

$$
\begin{aligned}
\frac{p(\hat{R}|X_{adj}, \bar{R} = \vec{1})}{p(\hat{R}|X_{adj}, \bar{R}_s = \vec{1})} &= p(\hat{R}|X_{adj}, \bar{R} = \vec{1}) \frac{p(\bar{R} = \vec{1}|X_{adj}, \hat{R}, \bar{R}_s = \vec{1})}{p(\hat{R}|X_{adj}, \bar{R}_s = \vec{1}, \bar{R} = \vec{1})p(\bar{R} = \vec{1}|X_{adj}, \bar{R}_s = \vec{1})} \\
&= \frac{p(\hat{R}|X_{adj}, \bar{R} = \vec{1})}{p(\hat{R}|X_{adj}, \bar{R} = \vec{1})} \frac{p(\bar{R} = \vec{1}|X_{adj}, \hat{R}, \bar{R}_s = \vec{1})}{p(\bar{R} = \vec{1}|X_{adj}, \bar{R}_s = \vec{1})} \\
&= \frac{p(\bar{R} = \vec{1}|X_{adj}, \hat{R}, \bar{R}_s = \vec{1})}{p(\bar{R} = \vec{1}|X_{adj}, \bar{R}_s = \vec{1})}
\end{aligned}
\tag{A.10}
$$

The second fraction is:

$$
\frac{p(X_{adj})}{p(X_{adj}|\bar{R}_s = \vec{1})} = \frac{p(\bar{R}_s = \vec{1})}{p(\bar{R}_s = \vec{1}|X_{adj})}
\tag{A.11}
$$

Combining both Eqs.(A.10) and (A.11) gives for the importance ratio:

$$
\begin{aligned}
\frac{p(\hat{R}|X_{adj}, \bar{R} = \vec{1})}{p(\hat{R}|X_{adj}, \bar{R}_s = \vec{1})} \frac{p(X_{adj})}{p(X_{adj}|\bar{R}_s = \vec{1})} &= \frac{p(\bar{R} = \vec{1}|X_{adj}, \hat{R}, \bar{R}_s = \vec{1})}{p(\bar{R} = \vec{1}|X_{adj}, \bar{R}_s = \vec{1})} \frac{p(\bar{R}_s = \vec{1})}{p(\bar{R}_s = \vec{1}|X_{adj})} \\
&= \frac{p(\bar{R} = \vec{1}|X_{adj}, \hat{R}, \bar{R}_s = \vec{1})p(\bar{R}_s = \vec{1})}{p(\bar{R} = \vec{1}|X_{adj})}
\end{aligned}
\tag{A.12}
$$

## A.8 IDENTIFICATION EXAMPLE FOR THE MODIFIED AFAIS

Here we show that the modified AFAIS estimator with $s = \{2\}$ is identified for the MNAR scenario from Appendix A.6 and that it is possible to evaluate the AFAIS estimator in this case. We follow the same example and see, first of all, that the propensity score now only needs to be evaluated on the data points where $\bar{R}_2 = 1$ which is possible. Secondly we show identification of the AFA propensity score:

**Identification of** $p(\bar{R} = \vec{1}|X_{adj}, \hat{R}, \bar{R}_s = \vec{1}) = p(\bar{R} = \vec{1}|X_1, X_2, \hat{R}, \bar{R}_2 = 1)$**:**
The identification of the AFA propensity score $p(\bar{R} = \vec{1}|X_1, X_2, \hat{R}, \bar{R}_2 = 1)$ is straight-forward, as the conditioning $\bar{R}_2 = 1$ makes this a function of observed data only.

**How to choose $s$:**
While we do not give general identifiability statements for the modified AFAIS estimator, we present here a simple guide for choosing the set $s$ that yields the most data efficient scenario under which the modified AFAIS can be used: First, start with $s = \emptyset$ (full off-environment bandit view) and investigate identifiability and the possibility of evaluation. If identification and evaluation is possible, then the entire dataset can be used for estimation. Otherwise, try larger sets for $s$ until identification and evaluation is possible or $s = \{0, \ldots, d_x\}$ has been reached. When the set $s = \{0, \ldots, d_x\}$ is reached (full black-box missing data view), then $J_{AFAIS}(s) \equiv J_{IPW}$ and the known identification theories for IPW hold.

## A.9 OTHER VIEWS ON AFAPE

### OFF-POLICY POLICY EVALUATION (OPE) FOR AFA

A completely different view on the AFAPE problem is off-policy policy evaluation (OPE). In OPE, one evaluates the performance of a target policy, the AFA policy, from data collected from a behavior policy which in this case corresponds to the retrospective missingness (e.g. the policy the doctors had in the hospital when deciding on what features to acquire). This view differs fundamentally from the AFA graph proposed in this paper in that it does not look at retrospective missingness as part of the environment, but rather as actions produced by a "behavior" policy. There are again two approaches that differ in their view on missingness. If both retrospective and AFA missingness are looked at from a sequence view (Figure A.3), the problem can be characterized as OPE of RL agents. One drawback of this view is that it requires additional knowledge about the order of acquisitions in

the retrospective dataset $\bar{U} = (\bar{U}^{(1)}, ...., \bar{U}^{(N)})$ which is often not known. An OPE alternative, is to look at retrospective and AFA missingness from a set perspective (with actions $\bar{R}/\hat{R}$) and therefore to consider the problem as OPE of bandit agents. This again uses the fact that the order of acquisitions does not have an impact on the evaluation step. In the following, we will, however, continue with the OPE RL perspective as this view has been applied to AFA before (Chang et al., 2019), but the described characteristics hold similarly also for the OPE bandit view.

For clarity, we denote the AFA missingness policy as $\pi_{AFA} \equiv \pi(\hat{U}^{(k)}|\hat{X}^{(k)}, \hat{R}^{(k)})$ and the retrospective missingness policy as $\pi_{Retro} \equiv \pi(\bar{U}^{(k)}|X, \bar{R}^{(k)})$. Note that $\bar{U}$ and $\hat{U}$ now describe the same variable, but the former is the retrospective distribution that acquired the data, while the latter is the interventional distribution that we would like to apply when deploying the AFA agent. A fundamental difference between the two distributions is, however, that $\pi_{Retro} \notin \mathcal{F}_{MAR}$ in general, which implies arrows $X \to \bar{U}^{(k')} \forall k'$ (depicted as blue arrows in Figure A.3) that are not mediated by $\bar{X}^{(k')}$.

The evaluation of $J$ under any AFA policy $\pi_{AFA}$ from the OPE RL view (and Assumption 1) corresponds to:

$$J_{do(\bar{R}=\vec{1})} = \mathbb{E}[C|do(\bar{R}=\vec{1})] = \mathbb{E}[C|do(\bar{U} \sim \pi_{AFA})] \equiv J_{do(\bar{U} \sim \pi_{AFA})} \tag{A.13}$$

This reads as: *the expected cost had, instead of the retrospective policy, the AFA policy been applied.* This shows that the AFAPE problem can be formulated using the AFA graph (with black-box missing data and off-environment bandit views) or using this alternative OPE view and that all views can be used to solve the same estimation problem. As the AFA policy does not depend on $X$, but only on $\hat{X}$, the arrows $X \to \bar{U}^{(k')} \forall k'$ are eliminated under the intervention $do(\bar{U} \sim \pi_{AFA})$. Thus, in this setting, $X$ is also a confounder due to for example the backdoor path $\bar{U}^{(k')} \leftarrow X \to Y \to \tilde{C}$. Therefore, unbiased evaluation requires adjustment of $X$ which is complex due to potentially missing $X$ values. However, adjustment is not needed under the assumption $\pi_{Retro} \in \mathcal{F}_{MAR}$, which means that the retrospective data was potentially also acquired by a *retrospective "AFA" agent* (e.g. the physician in the hospital). A huge drawback of the OPE view (from both RL and bandit perspectives) is that it is very data inefficient as we show in Appendix A.10.

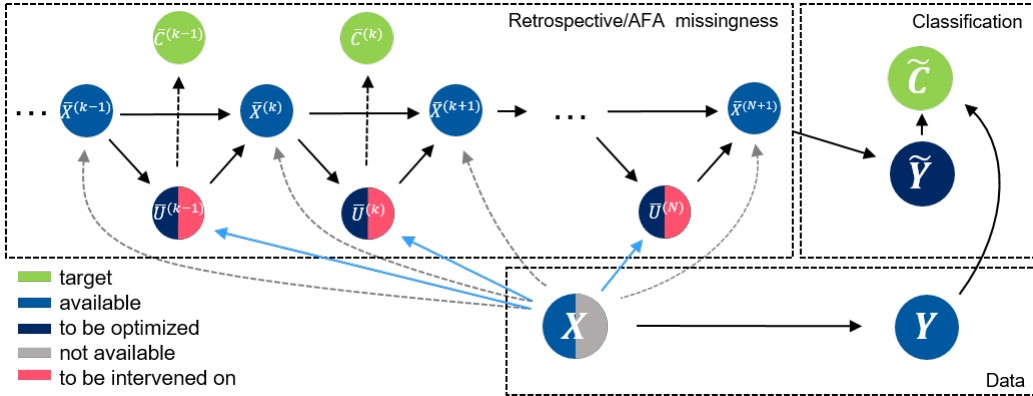

Figure A.3: AFA setting from an OPE RL view. The graph describes the observational data given by the retrospective missingness policy $\pi_{Retro} = \pi(\bar{U}^{(k)}|X, \bar{R}^{(k)})$. The state (shown as $\bar{X}^{(k)}$) also contains $\bar{R}^{(k)}$ which was left out for clarity. The run-time application of AFA agents then corresponds to the intervention $do(\bar{U} \sim \pi_{AFA})$ in which arrows $X \to \bar{U}$ are eliminated. This necessitates adjustment of the confounder $X$.

## BLOCKING ACQUISITION ACTIONS

The last common approach used to handling missing data in AFA settings is to block a feature acquisition action if the corresponding feature is missing (Janisch et al., 2020; Yoon et al., 2018). Yoon et al. (2018) additionally supplied the AFA agent with the missingness indicators in the retrospective

dataset, making use of "informative missingness". This approach, however, corresponds to evaluating $J_{\bar{R}} \equiv \mathbb{E}[C]$ and not $J_{do(\bar{R}=\vec{1})}$ and thus is a biased estimator in the setting we consider (due to Assumption 1). Under Assumption 1, there are no "informative missingness" indicators at run-time that could be given to the agent. This method is unbiased only under the alternative assumption that the exact same restrictions for the agent at train/evaluation time apply also at run-time.

## A.10 DATA EFFICIENCY AND POSITIVITY ASSUMPTIONS OF DIFFERENT VIEWS ON AFAPE

In this work, we propose different views on the retrospective and the AFA missingness. Here, we derive first the data efficiency of importance sampling approaches of the different views for the AFA setting; and then show the resulting implications of the required positivity assumptions of each approach.

### DATA EFFICIENCY

We represent data efficiency for AFA by two key quantities: 1) $n_{\hat{U}}$: the number of distinct AFA sequence trajectories $\hat{U}$ that can be evaluated using a single retrospective data point (defined by $Y$, $\bar{X}$, $\bar{R}$ and $\bar{U}$), and 2) $n_{\text{runs}}$: the number of times we have to run an agent on the data point (i.e. choose the data point as the environment) to be able to evaluate these trajectories. Table A.10 presents the views on retrospective and AFA missingness of different approaches, along with their data efficiency. We present the details of derivation and conclusion for each approach below.

We start with a short example of a data point with 3 features with given missingness $\bar{u} =$ ("observe($X_3$)", "observe($X_1$)", "stop observation") / $\bar{r} = (1, 0, 1)$ before deriving the general data efficiency for each view. We abbreviate with a slight abuse of notation: $\bar{u} = (3, 1)$.

| **Approach** | **Retrospective missingness** | **AFA missingness** | **Data efficiency** |
|---|---|---|---|
| *OPE RL view* | Sequence view | Sequence view | $n_{\hat{U}} = n_{\text{runs}} = 1$ |
| *OPE bandit view* | Set view | Set view | $n_{\hat{U}} = \|\bar{R}\|_1!$ 
 $n_{\text{runs}} = 1$ |
| *Black-box missing data view* | Set view (as environment) | Black-box function | If $\bar{R} = \vec{1}$ : $n_{\hat{U}} = \sum_{i=0}^{d_x} i!\binom{d_x}{i}$ 
 $n_{\text{runs}} = 2^{d_x}$ 
 If $\bar{R} \neq \vec{1}$ : $n_{\hat{U}} = n_{\text{runs}} = 0$ |
| *Off-environment bandit view* | Set view (as environment) | Set view | $n_{\hat{U}} = \sum_{i=0}^{\|\bar{R}\|_1} i!\binom{\|\bar{R}\|_1}{i}$ 
 $n_{\text{runs}} = 2^{\|\bar{R}\|_1}$ |
| *Off-environment RL view* | Set view (as environment) | Sequence view | $n_{\hat{U}} = n_{\text{runs}} = \sum_{i=0}^{\|\bar{R}\|_1} i!\binom{\|\bar{R}\|_1}{i}$ |

Table A.1: Different approaches on the AFAPE problem with their corresponding views on retrospective and AFA missingness, and the data efficiency. $n_{\hat{U}}$ denotes the number of AFA trajectories that can be evaluated from a single retrospective data point using the respective approach. $n_{\text{runs}}$ denotes the number of times an agent has to be run on the datapoint to evaluate the $n_{\hat{U}}$ trajectories. The off-environment bandit view is strictly more data efficient than all other views.

**Off-policy policy evaluation (OPE) view (RL perspective)**
In the OPE view from an RL perspective, the retrospective sequence trajectory $\bar{u} = (3, 1)$ is used directly to evaluate the target trajectory $\hat{u} = (3, 1)$. Thus, in the example and in general, one data point can be used to evaluate one target trajectory ($n_{\hat{U}} = 1$) with one run ($n_{\text{runs}} = 1$). This view is thus very data inefficient. It has the fundamental drawback that only trajectories of the behavior

policy are used to evaluate the target policy. It does not make use of the fact that one can also assess what would happen if less features were used or if the features were acquired in a different order.

**OPE view (bandit perspective)**
In the OPE view from a bandit perspective, the retrospective missingness indicator $\bar{r} = (1, 0, 1)$ is treated as one action (set view). It is then used to evaluate the target action $\hat{r} = (1, 0, 1)$. This jointly evaluates the AFA sequence trajectories $\hat{u} \in \{(1, 3), (3, 1)\}$. Thus, the exemplary data point can be used to simultaneously evaluate two target sequence trajectories ($n_{\hat{U}} = 2$) with one run ($n_{\text{runs}} = 1$). The number of trajectories that can be evaluated in general is equal to the number of permutations of the retrospectively-acquired features, i.e. $n_{\hat{U}} = \|\bar{R}\|_1!$, where the $L1$ norm $\|\bar{R}\|_1$ gives the number of acquired features.

This view is similar to the OPE view from the RL perspective shown above, except that it leverages the fact that the order of acquisitions does not have an influence on evaluation. Therefore, it considers all possible permutations which result the final set of features. Otherwise, it shares the same disadvantages as the OPE view from an RL perspective.

**Black-box missing data view**
In the black-box missing data view, the retrospective missingness is seen as part of the environment and the agent is allowed to choose different combinations of the available features. Only fully observed data points can, however, be used for evaluation. Therefore, one can evaluate no AFA sequence trajectories for the exemplary data point as it is not fully observed. Consider instead a (fully-observed) data point with retrospective missingness $\bar{u}^* = (3, 1, 2)$, $\bar{r}^* = (1, 1, 1)$. This data point can be used to evaluate all possible AFA sequence trajectories $\hat{u}^* \in \{(1, 2, 3), (1, 3, 2), ..., (1, 2), .., (1), (2), (3), ()\}$ (16 distinct trajectories). All trajectories that share the same AFA missingness ($\hat{r}^*$) can be simultaneously evaluated (as in the bandit view), thus requiring one run per possible subset, i.e. $\hat{r}^* \in \{(1, 1, 1), (1, 1, 0), ..., (1, 0, 0), ..., (0, 0, 0)\}$ (8 distinct sets). In general, one obtains $n_{\hat{U}} = \sum_{i=0}^{d_x} i! \binom{d_x}{i}$ and $n_{\text{runs}} = 2^{d_x}$ if $\bar{R} = \vec{1}$ ($d_x$ is the total number of features).

The black-box missing data view has the advantage of being a straight-forward approach based on well-established literature. The downside is that it treats the AFA policy as any other function applied to the data, and rejects the chance of evaluating the agent on incomplete data if the missing features is not desired by the agent in the first place. This view is only recommended in complex missing data scenarios where AFAIS fails, namely where the fraction of missing values is very low or when the AFA policy is "data hungry", i.e. the agent often acquires almost all the features.

**Off-environment bandit view**
Like the black-box missing data view, the off-environment bandit view also treats the retrospective missingness as part of the environment. The difference is that partially-observed data points can still be used for evaluation. In particular, all AFA sequence trajectories that contain a subset of the available features can be evaluated. For our example, the trajectories are $\hat{u} \in \{(1, 3), (3, 1), (1), (3), ()\}$ (5 distinct trajectories) which require 4 runs, i.e. one run per unique set ($\bar{r} \in \{(1, 0, 1), (1, 0, 0), (0, 0, 1), (0, 0, 0)\}$). In general, one can evaluate $n_{\hat{U}} = \sum_{i=0}^{\|\bar{R}\|_1} i! \binom{\|\bar{R}\|_1}{i}$ different sequence trajectories with $n_{\text{runs}} = 2^{\|\bar{R}\|_1}$ different runs.

This view combines the advantages of the OPE view (partially-observed data points can be used) and the missing data view (one can evaluate trajectories with subsets of the available features). It is strictly more data efficient than the OPE views and the black-box missing data view.

**Off-environment RL view**
The off-environment RL view also treats the retrospective missingness as part of the environment, but looks at the AFA missingness from a sequence view. This view thus also allows the evaluation of $n_{\hat{U}} = \sum_{i=0}^{\|\bar{R}\|_1} i! \binom{\|\bar{R}\|_1}{i}$ different sequence trajectories, but requires as many runs $n_{\text{runs}} = n_{\hat{U}}$. This view is equal to the off-environment bandit view, with the exception that it does not make use of the fact that the order of acquisitions doesn't have an influence on evaluation. It is thus sub-optimal compared to the off-environment bandit view.

POSITIVITY ASSUMPTIONS

Importance sampling approaches have to fulfill a positivity assumption that requires the behavior distribution to have positive support wherever the target distribution has positive support. Since behavior and target distributions differ between the different views, the required positivity assumptions will also be different. We first look at general positivity assumptions:

**General positivity assumptions**
In order to evaluate *any* possible AFA policy, the following positivity assumptions must hold:

*OPE view (RL perspective):* $p(\bar{U} = u|X) > 0 \,\forall X, u \in \mathcal{U}$ where $\mathcal{U}$ is the set of all possible sequences of actions
The positivity assumption of the OPE view requires each exact trajectory of actions to have positive support (for all possible feature values). It is the strongest of the positivity assumption in this list.

*OPE view (bandit perspective):* $p(\bar{R} = r|X) > 0 \,\forall X, r \in \{0,1\}^{d_x}$
The OPE view from a bandit perspective requires only each subset of acquired features to have positive support (for all possible feature values).

*Black-box missing data view:* $p(\bar{R} = \vec{1}|X) > 0 \,\forall X$
The black-box missing data view requires the possibility of a complete case (for each possible feature value). This is a much weaker assumption than the assumptions for the OPE views.

*Off-environment bandit/RL view:* $p(\bar{R} = \vec{1}|X) > 0 \,\forall X$
The off-environment bandit view has the same positivity assumption as the black-box missing data view when the goal is to evaluate any possible AFA policy.

Note that these are minimal positivity requirements. A complex missing data scenario might impose additional positivity assumptions for each of these views to allow identification. In the missing data view, for example, to identify any identifiable missing data scenario that can be represented by a DAG (directed acyclic graph), the following additional requirement must be met (Malinsky et al., 2020): $p(\bar{R} = \vec{1} - \vec{e_i}|X) > 0 \,\forall X, i$ s.t. $i$ indexes a partially missing feature. $\vec{e_i}$ denotes a vector of zeros with a 1 at position $i$.

**AFA policy specific positivity assumptions**
The general positivity assumptions will relax to some extent, if we are interested in evaluating specific AFA policies only with known support. In this case, the following modifications are to be made:

*OPE view (RL perspective):* $p(\bar{U} = u|X) > 0 \,\forall X, u$ s.t. $p(\hat{U} = u|X) > 0$
The OPE view from an RL perspective thus requires the behavior policy to only have positive support for action trajectories that could be taken by the AFA policy.

*OPE view (bandit perspective):* $p(\bar{R} = r|X) > 0 \,\forall X, r$ s.t. $p(\hat{R} = r|X) > 0$
Similarly, the OPE view from a bandit perspective requires the behavior distribution to only have positive support for sets of features that could be acquired by the AFA agent.

*Black-box missing data view:* $p(\bar{R} = \vec{1}|X) > 0 \,\forall X$
The missing data view treats the AFA missingness as a black-box function and can thus not exploit the scenario where we look at only specific AFA policies. It thus still has the same positivity assumption.

*Off-environment bandit/RL view:* $p(\bar{R} \in \{r' : r_i' \geq r_i \forall i\}|X) > 0 \,\forall X, r$ s.t. $p(\hat{R} = r|X) > 0$
The off-environment view requires the behavior distribution to have positive support for a superset of the set of features that could be acquired by the AFA agent. This is thus strictly the weakest positivity assumption amongst all views.

A.11 EXPERIMENT DETAILS

In this section, we first describe the general experiment and training setup before explaining each dataset in detail. Afterwards, we provide a detailed list of the parameters and configurations for each experiment.

**Data, costs and synthetic missing data**

For the experiments, we defined a "superfeature" as a feature that comprises multiple subfeatures, which are usually acquired or skipped jointly and which have a single cost (e.g. the pixels of an image). Furthermore, we assumed a subset of features is available at no cost (free features) and generated random acquisition costs $c_{acq}$ for the remaining features (from a uniform distribution between 0.5 and 1). We simulated misclassification costs to force good policies to find a balance between feature acquisition cost and predictive value of the features. We induced synthetic missingness with MCAR, MAR and MNAR missingness mechanisms similar to the scenarios described in Appendices A.6 and A.8. The distribution of the missingness indicators follows the logistic model (e.g. $p(\bar{R}_2 = 1|X_1) = \sigma(-X_1 + 1)$). To evaluate convergence of different estimators, we consider the average cost of running the AFA agent on the dataset over all data points in the ground truth testset (without missingness) as the true expected cost ($J_{do(\bar{R}=\vec{1})}$).

**Training**

We used an Impute-then-regress classifier (Le Morvan et al., 2021) with unconditional mean imputation and a random forest classifier for the classification task and trained it on fully available and randomly subsampled data (where $p(\hat{R}_i = 1) = 0.5$). We tested three random acquisition policies that acquire each costly feature with 10%, 50% and 90% probability, respectively. Furthermore, we evaluated a vanilla Deep Q-Network (DQN) RL agent (Mnih et al., 2015) which during training, blocks feature acquisitions if the respective feature is not available in the retrospective data. The nuisance models (propensity score and afa propensity score) were trained on the test set using multi-layer perceptrons and a cross-fitting approach was employed in the real-world data experiments to make maximum use of the data (Kennedy, 2022). In cross-fitting, we split the data into $K$ disjoint folds and train $K$ different sets of nuisance models where nuisance model $i$ is trained on all folds except fold $i$. The estimator is then evaluated on all folds by using the nuisance models $i$ if a data point is in fold $i$. We used $K = 2$ in all real-world experiments.

**Synthetic dataset**

We evaluated and compared the described methods on a synthetic dataset with $X \in \mathbb{R}^4$ and a binary label $Y$, simulated using Scikit-learn's `make_classification` function (Pedregosa et al., 2011). We assumed one superfeature "super$X_3$" to include $X_3$ and $X_4$. The synthetic data experiment contains MNAR missingness for which we assumed *i)* $X_1$ (fully observed) causes missingness in $X_2$; *ii)* $X_2$ causes missingness in super$X_3$. We chose $s = \{1\}$ as the conditioning set when using the modified AFAIS estimator to allow evaluation. The fraction of complete cases is $p(\bar{R} = \vec{1}) \approx 12\%$ in this experiment. For the synthetic data experiment we did not use cross-fitting, but instead generated a separate datasplit (40'000 data points) to train the nuisance models (See Table A.2 for full details).

**HELOC dataset**

We also tested on the HELOC (Home Equity Line of Credit) dataset which is part of the FICO explainable machine learning (xML) challenge (FICO, 2018). The dataset contains credit applications made by homeowners. The ML task was to predict based on information at the application, whether an applicant is able to repay their loan within two years. The dataset contains 9 superfeatures (See Table A.3 for full details).

**Retinopathy dataset**

Next, we tested on the Retinopathy dataset (Antal & Hajdu, 2014). The ML task was to predict diabetic retinopathy based on extracted image features. We considered image quality assessment and pre-screening as free features and manually extracted image features by an expert as costly. The dataset contains 7 superfeatures (See Table A.4 for full details).

**Adult income dataset**

We performed evaluation on the Income dataset from the UCI data repository (Newman et al., 1998). The ML task was to predict whether a person has an income over 50'000$. We considered age as a free feature and other private information (such as education, work experience, etc.) as costly features. The dataset contains 10 superfeatures (See Table A.5 for full details).

| Data and environment | |
|---|---|
| Sample size $n_{D_{\text{test}}}$ | 10′000 (plus separate datasplit of $n_{D_{\text{nuisance}}} = 40′000$ for training of the propensity and AFA propensity scores) |
| Superfeatures | super$X_0$: $[X_0]$, super$X_1$: $[X_1]$, super$X_2$: $[X_2, X_3]$ |
| Label | Y (class 0: 50%, class 1: 50%) |
| Feature acquisition cost | $c_{acq} = [0.0, 0.84, 0.86]$ |
| Misclassification cost | $c_{mc}(0,1) = c_{mc}(1,0) = 10.0$ |

| Missingness mechanisms | |
|---|---|
| MNAR | $p(R_0 = 1) = 1.0,$ 
 $p(R_1 = 1) = \sigma(1.0 - X_0),$ 
 $p(R_2 = 1) = \sigma(-1.0 - X_1)$ 
 Complete cases ratio: $p(\bar{R} = \vec{1}) = 0.12$ |

| Models | |
|---|---|
| Classifier | RandomForest (number of estimators: 5, max depth: 3) |
| Agents | Random (10%, 50%, 90%) 
 DQN (learning rate: 0.001, number of layers: 3, 
 hidden layer neurons: 50, hidden layer activation function: ReLU) |
| Nuisance functions | PS (number of layers: 2, hidden neurons per layer: 32, epochs: 2000, 
 hidden layer activation function: ReLU) 
 AFAPS (number of layers: 3, hidden neurons per layer: 32, epochs: 8000, 
 hidden layer activation function: ReLU) |

Table A.2: Full experiment details: Synthetic dataset

| Data and environment | |
|---|---|
| Sample size $n_{D_{\text{test}}}$ | 35562 (data split: 50% train, 50% test) |
| Superfeatures | super$X_0$: ['ExternalRiskEstimate'] |
| | super$X_1$: ['MSinceOldestTradeOpen', 'MSinceMostRecentTradeOpen', 'AverageMInFile', 'NumSatisfactoryTrades'] |
| | super$X_2$: ['NumTrades60Ever2DerogPubRec', 'NumTrades90Ever2DerogPubRec'] |
| | super$X_3$: ['PercentTradesNeverDelq', 'MSinceMostRecentDelq', 'MaxDelq2PublicRecLast12M'] |
| | super$X_4$: ['MaxDelqEver', 'NumTotalTrades'] |
| | super$X_5$: ['NumTradesOpeninLast12M', 'PercentInstallTrades', 'MSinceMostRecentInqexcl7days'] |
| | super$X_6$: ['NumInqLast6M, NumInqLast6Mexcl7days'] |
| | super$X_7$: ['NetFractionRevolvingBurden', 'PercentTradesWBalance'] |
| | super$X_8$: ['NetFractionInstallBurden', 'NumRevolvingTradesWBalance', 'NumInstallTradesWBalance', 'NumBank2NatlTradesWHighUtilization'] |
| Label | 'RiskPerformance' (class 0: 48%, class 1: 52%) |
| Feature acquisition cost | $c_{acq} = [0.56, 0.84, 0., 0.58, 0.6, 0.6, 0.64, 0.57, 0.54]$ |
| Misclassification cost | $c_{mc}(0,1) = c_{mc}(1,0) = 15.0$ |
| Missingness mechanisms | |
| MCAR | $p(R_i = 1) = 1.0, \quad i \in \{0, 2, 4, 6, 8\}$ 
 $p(R_j = 1) = 0.5, \quad j \in \{1, 3, 5, 7\}$ 
 Complete case ratio: $p(\vec{R} = \vec{1}) = 0.06$ |
| MAR | $p(R_i = 1) = 1.0, \quad i \in \{0, 2, 4, 6, 8\}$ 
 $p(R_j = 1) = \sigma(0.0 + 3.0\ \text{ExternalRiskEstimate})$ 
 Complete case ratio: $p(\vec{R} = \vec{1}) = 0.21$ |
| MNAR | $p(R_0 = 1) = 0.8,$ 
 $p(R_i = 1) = 1.0, \quad i \in \{2, 4, 6, 8\}$ 
 $p(R_j = 1) = \sigma(0.0 + 3.0\ \text{ExternalRiskEstimate})$ 
 Complete case ratio: $p(\vec{R} = \vec{1}) = 0.17$ |
| Models | |
| Classifier | RandomForest (number of estimators: 100, max depth: 15) |
| Agents | Random (10%, 50%, 90%) 
 DQN (learning rate: 0.001, number of layers: 3, hidden layer neurons: 50, hidden layer activation function: ReLU) |
| Nuisance functions | PS (number of layers: 2, hidden neurons per layer: 32, epochs: 2000, hidden layer activation function: ReLU) 
 AFAPS (number of layers: 3, hidden neurons per layer: 32, epochs: 8000, hidden layer activation function: ReLU) |

Table A.3: Full experiment details: HELOC dataset

| Data and environment | |
|---|---|
| Sample size $n_{D_{\text{test}}}$ | 1151 (data split: 50% train, 50% test) |
| Superfeatures | super$X_0$: ['quality assessment'] 
 super$X_1$: ['pre-screening'] 
 super$X_2$: ['MA detection {i}'],    i= $\{1, ..., 6\}$ 
 super$X_3$: ['exudates_{i}'],    i= $\{1, ..., 8\}$ 
 super$X_4$: ['dist macula opt disc'] 
 super$X_5$: ['diam opt disc'] 
 super$X_6$: ['class AM FM'] |
| Label | 'DR' (class 0: 47.0%, class 1: 53.0%) |
| Feature acquisition cost | $c_{acq} = [0., 0., 0.55, 0.57, 0.78, 0.52, 0.64]$ |
| Misclassification cost | $c_{mc}(0,1) = c_{mc}(1,0) = 12.0$ |
| Missingness mechanisms | |
| MCAR | $p(R_i = 1) = 1.0, \quad i \in \{0, 1, 6\}$ 
 $p(R_j = 1) = 0.7, \quad j \in \{2, 3, 4, 5\}$ 
 Complete case ratio: $p(\bar{R} = \vec{1}) = 0.23$ |
| MAR | $p(R_i = 1) = 1.0, \quad i \in \{0, 1, 6\}$ 
 $p(R_j = 1) = \sigma(-1.0 + 2.0 \text{ pre-screening}), \quad j \in \{2, 3, 4, 5\}$ 
 Complete case ratio: $p(\bar{R} = \vec{1}) = 0.28$ |
| MNAR | $p(R_0 = 1) = p(R_6 = 1) = 1.0, \quad p(R_6 = 1) = 0.9$ 
 $p(R_j = 1) = \sigma(-1.0 + 2.0 \text{ pre-screening}), \quad j \in \{2, 3, 4, 5\}$ 
 Complete case ratio: $p(\bar{R} = \vec{1}) = 0.22$ |
| Models | |
| Classifier | RandomForest (number of estimators: 40, max depth: 6) |
| Agents | Random (10%, 50%, 90%) 
 DQN (learning rate: 0.001, number of layers: 3, 
      hidden layer neurons: 50, hidden layer activation function: ReLU) |
| Nuisance functions | PS (number of layers: 2, hidden neurons per layer: 32, epochs: 2000, 
      hidden layer activation function: ReLU) 
 AFAPS (number of layers: 3, hidden neurons per layer: 32, epochs: 4000, 
      hidden layer activation function: ReLU) |

Table A.4: Full experiment details: Retinopathy dataset

| **Data and environment** | |
|---|---|
| Sample size $n_{D_{\text{test}}}$ | 32561 (data split: 50% train, 50% test) |
| Superfeatures | workclass: ['Federal-gov', 'Local-gov', 'Never-worked', 'Private', 'Self-emp-inc', 'Self-emp-not-inc', 'State-gov', 'Without-pay'] 

 education: ['1st-4th', '5th-6th', '7th-8th', '9th', '10th', '11th', '12th', 'Assoc-acdm', 'Assoc-voc', 'Bachelors', 'Doctorate', 'HS-grad', 'Masters', 'Preschool', 'Prof-school', 'Some-college', '-num'] 

 marital-status: ['Married-AF-spouse', 'Married-civ-spouse', 'Married-spouse-absent', Never-married', 'Separated', 'Widowed', 'relationship Not-in-family', 'relationship Other-relative', 'relationship Own-child', 'relationship Unmarried', 'relationship Wife'] 

 occupation: ['Adm-clerical', 'Armed-Forces', 'Craft-repair', 'Exec-managerial', 'Farming-fishing', 'Handlers-cleaners', 'Machine-op-inspct', 'Other-service', 'Priv-house-serv', 'Prof-specialty', 'Protective-serv', 'Sales', 'Tech-support', 'Transport-moving'] 

 race: ['Asian-Pac-Islander', 'Black', 'Other'], sex: ['sex Male'] 
 age: ['age'], hours-per-week: ['hours-per-week'] 
 capital-gain: ['capital-gain'], capital-loss: ['capital-loss'] |
| Label | 'income' (class 0: 76.0%, class 1: 24.0%) |
| Feature acquisition cost | $c_{acq} = [0.53, 0.64, 0.64, 0.53, 0.64, 0.55, 0., 0.82, 0.52, 0.76]$ |
| Misclassification cost | $c_{mc}(0, 1) = c_{mc}(1, 0) = 20.0$ |
| **Missingness mechanisms** | |
| MCAR | $p(R_i = 1) = 0.8, \quad i \in \{0, 2, 3, 4, 7, 8, 9\}$ 
 $p(R_j = 1) = 1.0, \quad j \in \{1, 5, 6\}$ 
 Complete case ratio: $p(\bar{R} = \vec{1}) = 0.16$ |
| MAR | $p(R_i = 1) = 1.0, \quad i \in \{5, 6\}$ 
 $p(R_j = 1) = \sigma(1.0 + \text{Male} + 4.0\,\text{age}), \quad j \in \{1, 2, 3, 4\}$ 
 $p(R_k = 1) = \sigma(1.0 + \text{Male} + 3.0\,\text{age}), \quad k \in \{7, 8, 9\}$ 
 Complete case ratio: $p(\bar{R} = \vec{1}) = 0.22$ |
| MNAR | $p(R_6 = 1) = 1.0, \quad p(R_5 = 1) = \sigma(2.0 + age),$ 
 $p(R_j = 1) = \sigma(1.0 + \text{Male} + 4.0\,\text{age}), \quad j \in \{1, 2, 3, 4\}$ 
 $p(R_k = 1) = \sigma(1.0 + \text{Male} + 3.0\,\text{age}), \quad j \in \{7, 8, 9\}$ 
 Complete case ratio: $p(\bar{R} = \vec{1}) = 0.20$ |
| **Models** | |
| Classifier | RandomForest (number of estimators: 100, max depth: 8) |
| Agents | Random (10%, 50%, 90%) 
 DQN (learning rate: 0.001, number of layers: 3, hidden layer neurons: 50, hidden layer activation function: ReLU) |
| Nuisance functions | PS (number of layers: 2, hidden neurons per layer: 32, epochs: 2000, hidden layer activation function: ReLU) 
 AFAPS (number of layers: 3, hidden neurons per layer: 32, epochs: 10000, hidden layer activation function: ReLU) |

Table A.5: Full experiment details: Income dataset

## A.12 ADDITIONAL EXPERIMENT RESULTS

Figure 5 in the main text shows the convergence of different estimators for the synthetic MNAR data experiment. The corresponding estimates are shown here in Figure A.4. The experiment further exemplifies how large the errors of biased estimators can be. Only $J_{IPW}$ and $J_{AFAIS}$ produce unbiased estimates for all agents. The estimates produced by $J_{Imp}$ and $J_{\bar{R}}$ are far from the ground truth $J_{do(\bar{R}=\vec{1})}$, especially for the DQN agent. This poses a high risk for model deployment, especially in safety-critical applications.

The main text also shows the estimates of the HELOC (MCAR scenario), Retinopathy (MAR scenario) and Income (MNAR scenario) real-world data experiments (Figure 6). We provide the experiment results for the remaining missingness scenarios in Figure A.5.

An additional benefit of IS-based estimators is that one can further use the estimator to not only estimate the counterfactual cost, but also other quantities of interest, such as counterfactual frequency of feature acquisitions. Figure A.6 shows

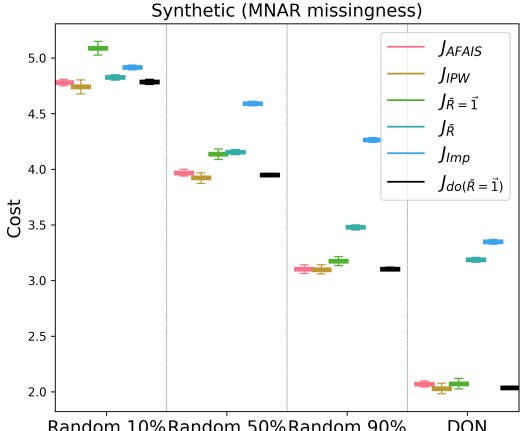

Figure A.4: Evaluated performance of different agents for the MNAR synthetic data experiment. Box plots shows variability of the estimate by bootstrapping 300 times without replacement 70% of a total test dataset of size $n_{D_{test}} = 100'000$.

the frequency of acquisitions for the DQN agent in MNAR Income dataset experiment. While the sampling is done on a dataset with missingness and blocked acquisition actions for non-available features (left plot), the AFAIS estimator (middle plot) gives estimates for the counterfactual frequency of acquisitions, i.e. how many times each feature would have been observed had the retrospective dataset been without missingness (right plot). As the result, the agent acquires the feature 'occupation' the most (among costly features) which is reasonable as it is a good indicator of income. The estimation of counterfactual frequency of acquisition decisions at run-time (e.g. in hospitals) can be a valuable information as it allows for resource management and allocation planning in advance.

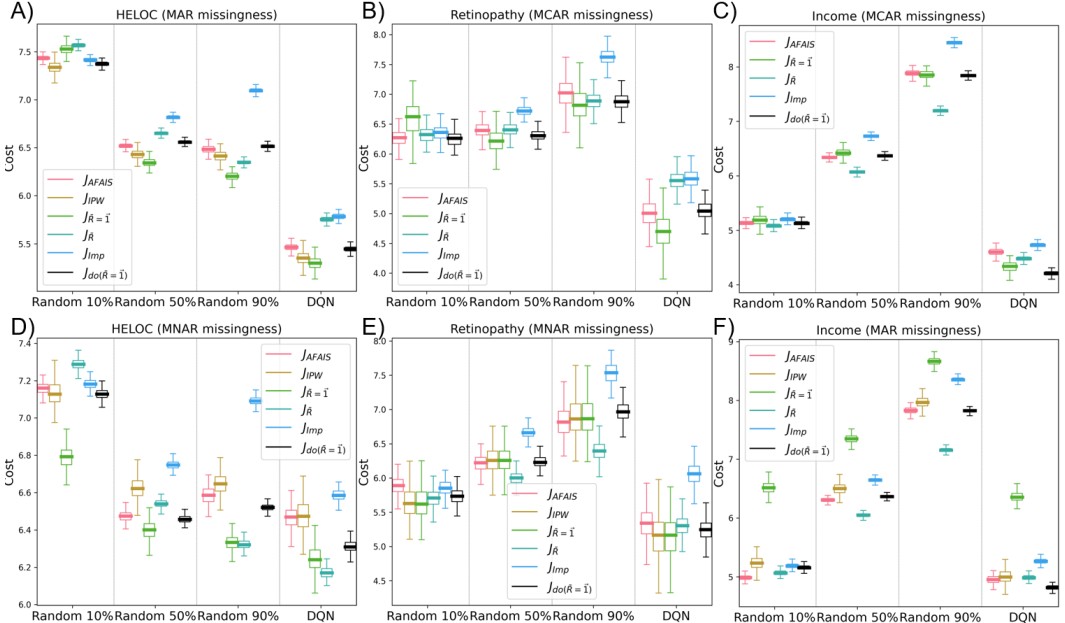

Figure A.5: Results for the HELOC, Retinopathy and Income dataset experiments for different missingness scenarios. This is a complement to Figure 6 in the main text and shows the remaining missing data scenarios (MAR/MNAR for HELOC, MCAR/MNAR for Retinopathy and MCAR/MAR for Income). Boxplots show variability of the estimate when bootstrapping 300 times without replacement 70% of a total test dataset of size $n_{D_{test}}$ ($n_{D_{test}} = 17'781$ for HELOC, $n_{D_{test}} = 575$ for Retinopathy and $n_{D_{test}} = 16'281$ for Income).

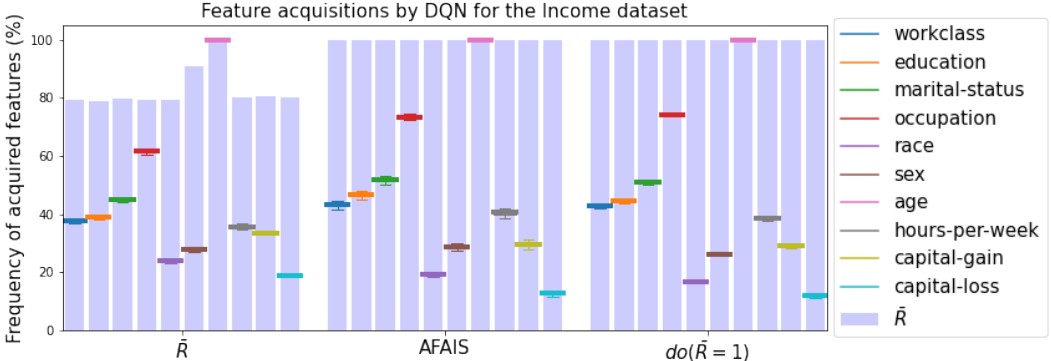

Figure A.6: Acquisitions of the DQN agent for the Income dataset. $\bar{R}$ corresponds to the acquisitions that were performed by the agent on the retrospective dataset. The acquisitions that were made by the agent when running it on the ground truth dataset (without missingness) are shown on the right ($do(\bar{R} = \vec{1})$). The middle shows the estimate by the AFAIS estimator.

## A.13 GLOSSARY OF TERMS AND SYMBOLS

| Term | Description |
|------|-------------|
| *run-time* | Time when the algorithm is applied prospectively (e.g. when deployed in the hospital) |
| *retrospective missingness* | Missingness in the retrospective dataset used to train/test the AI systems |
| *AFA missingness* | Missingness that is created at run-time when the AFA agent decides on feature acquisitions |
| *MCAR* | Missing completely at random |
| *MAR* | Missing at random |
| *MNAR* | Missing not at random |
| *AFAPE* | Active feature acquisition performance evaluation: The problem of estimating how an AFA agent and classifier would perform at run-time from data with retrospective missingness |
| *sequence view* | View on missingness with a focus on the sequence of feature acquisitions |
| *set view* | The traditional view on missingness with a focus on which set of features was acquired |
| *AFA graph* | Causal graph describing the AFA problem |
| *black-box missing data view* | View on AFAPE where retrospective missingness is seen from a set view and as part of the environment and AFA missingness is seen as a black-box function |
| *off-environment bandit view* | View on AFAPE where retrospective missingness is seen from a set view and as part of the environment and AFA missingness is also seen from a set view |
| *off-environment RL view* | View on AFAPE where retrospective missingness is seen from a set view and as part of the environment and AFA missingness is seen from a sequence view |
| *OPE* | Off-policy policy evaluation |
| *OPE view (RL perspective)* | View on AFAPE with a sequence view on both retrospective and AFA missingness |
| *OPE view (bandit perspective)* | View on AFAPE with a set view on both retrospective and AFA missingness |
| *IPW* | Inverse probability weighting |
| *MI* | Multiple imputation |
| *AFAIS* | Active feature acquisition importance sampling: novel estimator for the AFAPE problem based on the off-environment bandit view |
| *IS* | Importance sampling |
| *propensity score* | $p(\bar{R} = \vec{1}\|X)$ (weighting term in the IPW and AFAIS estimators) |
| *AFA propensity score* | $p(\bar{R} = \vec{1}\|X, \hat{R}, \bar{R}_s = \vec{1})$ (weighting term in the AFAIS estimators) |
| *nuisance function* | Function that needs to be fitted on the data in order to use a corresponding estimator (examples are the propensity score and AFA propensity score) |

| Symbol | Description |
|---|---|
| $X$ | Ground truth value of features (potentially unknown) |
| $Y$ | Ground truth value of the label |
| $\bar{R}$ | Missingness indicator (retrospective missingness) |
| $\bar{X}$ | Observed proxy for $X$ (retrospective missingness) |
| $\bar{U}^{(k)}$ | Feature acquisition action at acquisition step $k$ (retrospective missingness) |
| $\hat{R}$ | Missingness indicator (AFA missingness) |
| $\hat{X}$ | Observed proxy for $X$ (AFA missingness) |
| $\hat{U}^{(k)}$ | Feature acquisition action at acquisition step $k$ (AFA missingness) |
| $\pi$ | Missingness policy, i.e. probability of taking a certain feature acquisition action |
| $\mathcal{F}_{MAR}$ | Functional space for missingness policies that restricts decisions of acquiring new features to be based only on previously acquired features |
| $c_{acq}$ | Vector of feature acquisition costs: element $i$ is the cost of acquiring feature $X_i$ |
| $c_{mc}(y_l, y_m)$ | Misclassification cost of predicting a label of class $y_l$ as belonging to class $y_m$. |
| $\tilde{Y} = f(\hat{R}, \hat{X})$ | Predicted value for label $Y$ using classifier $f$ from features that have been acquired by the AFA agent |
| $C = \hat{C} + \tilde{C}$ | Random variable describing the total cost consisting of the accumulated feature acquisition $\hat{C}$ and the misclassification cost $\tilde{C}$ |
| $J = \mathbb{E}\left[C\right]$ | Expected cost of applying an AFA missingness policy $\pi$ and a classifier $f$ on a dataset with retrospective missingness |
| $J_{do(\bar{R}=\vec{1})} = \mathbb{E}\left[C \vert do(\bar{R} = \vec{1})\right]$ | Expected cost of applying an AFA missingness policy $\pi$ and a classifier $f$ at run-time (without retrospective missingness) |
| $h(X, Y) := \mathbb{E}[C \vert X, Y]$ | Expected cost given all features and the label |
| $J_{AFAIS}$ | Active feature acquisition importance sampling (AFAIS) estimator for $J_{do(\bar{R}=\vec{1})}$ |
| $J_{IPW}$ | Inverse probability weighting (IPW) estimator for $J_{do(\bar{R}=\vec{1})}$ |
| $J_{Imp}$ | Imputation-based estimator for $J_{do(\bar{R}=\vec{1})}$ |
| $J_{\bar{R}}$ | Estimator for $J_{do(\bar{R}=\vec{1})}$ based on the blocking of acquisition actions for not available features |
| $J_{\bar{R}=\vec{1}}$ | Estimator for $J_{do(\bar{R}=\vec{1})}$ that only considers those data points where all features are available (complete case analysis) |
| $s \subseteq \{0, \ldots, d_x\}$ | Hyperparameter of the modified AFAIS estimator that denotes which subset of the features must be complete (i.e. $\bar{R}_s = \vec{1}$) in order for a data point to be used in the evaluation. A smaller set $s$ implies higher data efficiency (less restrictions on complete information), while a larger set $s$ increases the possibility of identification and evaluation for a particular missingness scenario. |

