# OpenReview forum: "Evaluation of Active Feature Acquisition Methods under Missing Data"
_ICLR.cc/2023/Conference — Submitted to ICLR 2023_

### Official Review · Reviewer_zfWY · 2022-10-17

**Confidence:** 3
**Correctness:** 3
**Technical Novelty And Significance:** 4
**Empirical Novelty And Significance:** 3
**Recommendation:** 6

**Clarity, Quality, Novelty And Reproducibility:**

The paper is really clear and the literature gives a good overview of the field (I would recommend including it in the main text). The method is well introduced.

The paper proposes a novel approach to tackle the problem of data acquisition. My only concern is about the assumption of non-sequentiality in the testing procedure.


**Strength And Weaknesses:**

This paper explores an important problem in machine learning, particularly relevant to healthcare applications. This work provides a clear formalisation of the problem.

However, the model makes a key assumption that weakens the paper: "the order of acquisitions does not matter for the evaluation step". In the medical context, a first test, if positive, might make all other tests unnecessary. So, could further clarify this assumption?

**Summary Of The Paper:**

This paper studies the problem of data acquisition at test time when the training data contains missing data. This work tackles the question: which testing policy should be performed to minimize the cost but maximize performance? The paper shows that traditional ways to handle this problem lead to biased results when training data presents non completely at random missingness. The authors propose to an unbiased estimator of the cost associated with a testing policy.

**Summary Of The Review:**

This work tackles a central issue in machine learning for healthcare. However, one of the central assumptions weakens the claims of the paper.

---

> ### Author Response · Authors · 2022-11-19
> **Response to review**
>
> We thank the reviewer for their valuable time reviewing and thoughtful feedback.
>
> Weakness 1: "the order of acquisitions does not matter for the evaluation step"
>
> We agree completely with the reviewer that the order of acquisition is highly important for the doctor/AFA policy to decide on which features to acquire. The policy might, after a first positiv test not continue with other measurements. Therefore we frame AFA as a sequential decision problem.
>
> The order of acquisitions does, however, not matter for the evaluation step. This can be easily seen by checking how the cost is calculated. The feature acquisition cost is invariant with respect to the order of acquisitions. Acquiring first feature A and then feature B produces the same cost as acquiring first feature B and then feature A. The same holds for the misclassification cost. The classifier uses features A and B in the same way, irrespective of the order in which they were acquired. Therefore the outcome of interest (cost) from a trajectory where first feature A and then feature B was acquired can equally be used to assess what would have happened if first feature B and then feature A was acquired.
>
> Action: We added the following sentence to Section 3.3.2: "Although AFA was defined as a sequential decision problem, the order
> of acquisitions does not influence the evaluation step as the acquisition and misclassification costs
> are invariant to the order of acquisitions."
> We've also added an additional Appendix (A.10) to derive the data efficiency in each view. This Appendix also clarifies the effect of the role of the order of acquisition for the evaluation step.
>
> Clarity 1: Including AFA related methods in the main body
>
> Thank you for this suggestion. We would like to include this part in the main body, but due to limited space and conflicting reviewer requests regarding what should be added to the main paper, we would leave this in the Appendix.

---

> > ### Author Response · Authors · 2022-12-02
> > **Response to Reviewer**
> >
> > We would like to kindly thank you again for your comments and feedback. In light of the upcoming discussion deadline, we would appreciate it very much if you could let us know whether all your questions are addressed. We are also more than happy to answer any further questions.

---

### Official Review · Reviewer_qb3R · 2022-10-19

**Confidence:** 3
**Correctness:** 4
**Technical Novelty And Significance:** 4
**Empirical Novelty And Significance:** 3
**Recommendation:** 8

**Clarity, Quality, Novelty And Reproducibility:**

The quality of the writing is high, the same goes for the clarity apart from the points mentioned above. The approach itself is novel as far as I can say due to a somewhat limited overview of the literature in that domain. Some training details seem to be missing in the appendix to allow for full reproducibility (e.g., which gradient descent method, activation functions in the neural nets,...).

**Strength And Weaknesses:**

The authors tackle an important task with a novel and principled approach.

One weakness of the paper is in the theoretical section 3. The diversity yet similarity of the notation requires close attention to follow. The authors provide a very helpful glossary in the appendix, however, it is never mentioned in the main text. Adding a reference there would allow for greater readability. The second weakness similarly relates to the structure of the paper. Given the length of Section 3, almost all, especially all experiments with real-world data, are hidden in the appendix.



**Summary Of The Paper:**

The authors consider the task of active feature acquisition, i.e., the requirement of an actor to choose whether and if so which features to acquire during run time, in the context of missing data during the training stage (in all three variants of missingness). Their proposal formulates the task as a causal graph and formulates it as a reinforcement learning problem. Their final estimator of the true cost function is then evaluated in several experimental settings.


**Summary Of The Review:**

A paper tackling an important task with a principled solution, whose weaknesses lie primarily in its structural presentation and less in its theoretical contribution.

---

> ### Author Response · Authors · 2022-11-19
> **Response to review**
>
> We thank the reviewer for their valuable time reviewing and thoughtful feedback.
>
> Weakness 1: Missing mention of the glossary
>
> Thank you for pointing this out.
>
> Action: We've added a reference in the main text (Section 3.1).
>
> Weakness 2: Experimental results in the appendix
>
> Action: We moved some of the real-world experiment results to the main body and extended the results and discussion sections.
>
> Reproducability: Lack of reproducability
>
> Thank you for pointing this out.
>
> Action: We've added an extensive set of experiment details in Appendix A.11 to allow reproducability.

---

### Official Review · Reviewer_vxt6 · 2022-10-22

**Confidence:** 3
**Correctness:** 3
**Technical Novelty And Significance:** 2
**Empirical Novelty And Significance:** 2
**Recommendation:** 6

**Clarity, Quality, Novelty And Reproducibility:**

- The paper is easy to read and clear (except the experimental section).
- I think the proposed framework is somewhat novel.
- Some details are missing for the experiments and many details are deferred to the appendix; thus, I cannot provide high score for the reproducibility of this paper.

**Strength And Weaknesses:**

Strength:
- The authors tackled an important problem, active feature acquisition. Especially, the authors focused on evaluation of AFA model with missing components which is highly practical problem.
- The proposed solution makes sense and have sample efficiency and generally applicable to any missing patterns.

Weakness:
- The experimental sections are weak. The datasets are too simple; thus, hard to say that the proposed evaluation method can be well generalized to the real-world datasets.
- In experiments, the authors only focused on one-time AFA. However, in the introduction, the authors mainly discussed about the sequential AFA.

**Summary Of The Paper:**

- The authors tackled an important problem called active feature acquisition (AFA) which is critical but under-explored.
- More specifically, the authors focused on the setting where the training data includes missing components which is also practical in the setting when active feature acquisition is necessary (measurement costs are high).
- The authors do not focus on proposing a new AFA model. Instead, the authors focused on evaluation of AFA models with the training data with missing components.
- The experimental results are promising, showing consistently better evaluation results in comparison to alternatives in both synthetic and real-world datasets.

**Summary Of The Review:**

1. Experiments on real-world data
- With real world data we cannot do experiments because the assumption 1 does not hold with logged data.
- So, we should state that the real-world data experiments are actually somewhat semi-synthetic data experiments.

2. IPW
- When we use IPW, one another way is computing the probability of the agent that selects the same subset of logged features.
- Then, use that as the propensity score to compute the unbiased estimator of the cost.
- In other words, we can do weighted average of the logged costs where weights come from the probability that our AFA agent can select the exact subset of logged policy.
- In that case, we do not need to worry about only utilizing the samples with complete cases and this can significantly improve the sample efficiency of the propensity method.

3. Small number of data points
- In many medical domains, there are only a small number of samples in the datasets.
- It would be great if the authors can provide the results with a small number of samples ranging from 0-5000 as well.

4. Number of features
- It seems like the authors used the synthetic data with 4 features.
- It would be better if the authors can provide the results with more features.
- Based on the introduction, the authors mentioned emergency care examples with many possible measurements. To make this result more realistic, it would be better to provide the results with at least 20 features.
- Note that all the real-world experiments have less than 10 features.

5. Completeness of the experiments
- As we know, the missing patterns are quite important in this paper.
- In that case, it would be good if the authors provide MCAR, MAR, MNAR settings for one synthetic data and all 3 real-world datasets.
- Currently, the authors only provide one set of experimental results.

---

> ### Author Response · Authors · 2022-11-19
> **Response to review**
>
> We thank the reviewer for their valuable time reviewing and thoughtful feedback.
>
> Weakness 1: Simplicity of the datasets
>
> We chose the respective datasets because they are fully observed and allow the simulation of missing data. Simulation is a popular method in missing data problems as the ground truth is not known in real datasets and the performance of estimators could thus not be compared in those datasets.
> Unfortunately, we are not aware of other public datasets that fit the non time-series setting of AFA and which could be used for benchmarking.
>
> Weakness 2: One-time AFA
>
> We are not sure what the reviewer means by one-time vs sequential AFA. Throughout the whole paper (also in the experiments), the acquisition decisions are performend sequentially. The agent always has the opportunity to observe a feature value before the next observation decision has to be performed. For evaluation, however, we take a bandit view (i.e. we view the AFA process as if all features had been acquired jointly). This is, however, only for evaluation purposes and represents no restriction on the AFA method used. This view leverages the fact that the order of acquisitions is irrelevant for the evaluation step.
>
> If the reviewer refers by one-time vs sequential AFA to the static vs time-series setting, we would like to clarify that we only look at the static setting (no time-series data). The decision process is, however, also sequential in the static setting.
>
> Summary of the review 1: Experiments on real-world data
>
> Thank you for pointing this out.
>
> Action: We've clarified in several places of the paper that the missingness patterns in our experiments were artificially induced by ourselves. See e.g. the abstract: "... under induced MCAR, MAR and MNAR missingness"
>
> Summary of the review 2: IPW
>
> This suggestion corresponds to a bandit perspective on the OPE view that we discuss in Appendix A.9. We gave an intuition on why this approach is extremely data inefficient, but hadn't quantified it. In the rebuttal we have therefore added an additional Appendix A.10 that derives the data efficiency for each view, showing the OPE view is extremely data inefficient compared to the off-environment bandit view.
>
> Action:
> We've rewritten Appendix A.9 to emphasize the two views on OPE (RL and bandit view). We have further included Appendix A.10 to derive the data efficiency for each view.
>
> Summary of the review 3: Small number of datapoints
>
> Please see experiment results for the Retinopathy dataset. It has a testset data size of $n_D = 575$.
>
> Summary of the review 4: Number of features
>
> Unfortunately, we were not able run an experiment with > 20 features for the rebuttal deadline, but would be happy to include such results for the camera-ready version of the paper.
>
> Summary of the review 5: Completeness of the experiments
>
> Action: We have included additional experiment results in the Appendix such that the paper now contains results for MCAR, MAR and MNAR missingness scenarios for each of the real-world datasets.

---

> > ### Author Response · Authors · 2022-12-02
> > **Response to Reviewer**
> >
> > We would like to kindly thank you again for your comments and feedback. In light of the upcoming discussion deadline, we would appreciate it very much if you could let us know whether all your questions are addressed. We are also more than happy to answer any further questions.

---

> > > ### Comment · Reviewer_vxt6 · 2022-12-06
> > > **Thank you for the detailed response**
> > >
> > > Thank you for your response to my reviews.
> > > Also, thank you for providing additional experiments in the revised manuscript.
> > > Some of the comments are well addressed by the authors (e.g., Completeness of the experiments).
> > > However, some of the questions still remain.
> > >
> > > The experiments section of this paper is still weak.
> > > - As we know, real-world datasets have a much larger number of features than 10.
> > > - In that point of view, it is unclear whether this method can be generalized to more realistic scenarios (as authors mentioned in the introduction as medical examples).
> > > - In addition, as I mentioned in the previous review, "in experiments", the authors only focused on one-time AFA which is also somewhat less practical.
> > >
> > > In that point of view, I am standing on my original score (6) after carefully considering the rebuttals.

---

> > > > ### Author Response · Authors · 2022-12-08
> > > > **Response to Reviewer**
> > > >
> > > > Thank you for the response.
> > > >
> > > > 1. Number of features:
> > > > Regarding the number of features, we would like to additionally clarify the distinction between free features, features and superfeatures. Free features are features without a cost and should always be observed. We do not consider it as a task for the AFA agent to decide on their acquisition. This reduces the dimensionality, also in many real-world settings.
> > > > Superfeatures are features that typically comprise multiple subfeatures that are acquired/missing jointly. This can be for example the pixels of an image or different laboratory values that are acquired jointly in for example a complete blood count. This reduces the dimension of the problem also strongly. For example, our experiments contain up to 59 features (Income dataset), but the use of superfeatures reduces this dimension.
> > > >
> > > > 2. One-time AFA:
> > > > In the previous response, we pointed out the ambiguity of the "one-time AFA" question, and therefore had to make a best guess about the reviewer's thought. Since our explanations have not eliminated the reviewer's concern, we hope we receive a clarification of what is meant by "one-time AFA", in order to be able to provide a suitable answer (and potentially paper modification) for it.

---

### Official Review · Reviewer_pwju · 2022-10-24

**Confidence:** 2
**Correctness:** 3
**Technical Novelty And Significance:** 2
**Empirical Novelty And Significance:** 2
**Recommendation:** 6

**Clarity, Quality, Novelty And Reproducibility:**

The paper appears to introduce a novel, first-of-its-kind approach to active feature acquisition. Unfortunately, the paper is not written with a general audience in mind, which limits its potential impact across other communities. Investing half a page on intuitively introducing all key AFA concepts would go a long way to address this issue. Also the paper over-emphasizes the method's detail (5 pages) and, within its main body, it barely touches on the empirical validation (one page on a single, synthetic domain; per the comments above, the experiments from A.10 should be brought within the main paper). The paper would also greatly benefit (in both clarity and significance) if it had an in-depth discussion of the empirical insights, with their strengths and weaknesses).


**Strength And Weaknesses:**

The paper introduces a novel approach to an important, real-world problem: active feature acquisition. It sets up the problem, provides a novel framework for solving it, and validates it empirically against existing approaches.

However, the paper could be significantly improved by
(i) a better use of the available space: in its current form, most experimental results (including all on the real-world datasets) are in the appendix. The authors should shrink the current "section 3" by at least two pages (from 5 to at most 3) and use the freed space to bring forward the experiments in the appendix and to thoroughly discuss insights on their significance and weaknesses.
(ii) the paper would greatly benefit from a motivating, real-world dataset. The three datasets in Appendix A-10 appear to be real-world, but the experimental setup is quite vague and confusing (did the author(s) chose arbitrary are the levels of missingness and miss-classification costs? if not, please explain).

**Summary Of The Paper:**

The paper introduces AFAPE, the first approach active feature acquisition performance evaluation under missing data. It also introduces AFAIS (i.e.,  active feature acquisition importance sampling) a novel estimator that is more efficient than existing approaches.

**Summary Of The Review:**

Overall, the proposed approach has clear potential, but the paper needs quite a bit of re-writing/re-organizing in order to clarify its impact. As most of the required information is already within the appendices, the comments in this review should quite easy and fast to address.

---

> ### Author Response · Authors · 2022-11-19
> **Response to review**
>
> We thank the reviewer for their valuable time reviewing and thoughtful feedback.
>
> Weakness 1: Organization
>
> Thank you for this suggestion. We have reduced the Methods section to incorporate real-world experiment results and a more in-depth discussion in the main body. Unfortunately, we only have the limited amount of 9 pages within the main body and partly opposing suggestions by the reviewers (reviewer dNWk) on which parts should be in the main body and were thus not able to reduce the Methods section quite to the length requested by the reviewer.
> We had to find a good balance and believe the remaining content in the Methods section is necessary to have a self-contained main body. For example, we find it necessary to include section "3.2 PROBLEM FORMULATION: ACTIVE FEATURE ACQUISITION (AFA)" in the main body, although this is more of a preliminary description of AFA problems and does not yet introduce the AFAPE problem setting. The reason we find it necessary is that AFA is not a well-established field and problem settings differ in the literature.
>
> Action:
> We've reduced the Methods section to extend the Results and Discussion. We included real-world experiments in the main body. The main body now contains a more in-depth explanation of the results. We have also extended the Discussion section to emphasize on the types of questions that matter when choosing an estimator.
>
> Weakness 2: Experimental setup
>
> The reviewer mentions a confusing setup regarding the real-world experiments.
>
> Action: We have now incorporated an extensive set of experiment details in Appendix A.11 to improve reproducability and clarity on experiment design.
>
> Weakness 3: Lack of intuitive introduction of key AFA concepts
>
> We agree with the reviewer that the paper can be improved with a section that intuitively describes the different concepts of our AFA approach. Unfortunately, due to the space constraints we are not able to include this in the main body.
>
> Action: We added Appendix A.10 to give a clearer understanding of the benefits of each viewpoint in terms of data efficiency. We hope this new discussion can provide an intuitive understanding of why the off-environment bandit view is to be preferred over the other views.

---

> > ### Author Response · Authors · 2022-12-02
> > **Response to Reviewer**
> >
> > We would like to kindly thank you again for your comments and feedback. In light of the upcoming discussion deadline, we would appreciate it very much if you could let us know whether all your questions are addressed. We are also more than happy to answer any further questions.

---

### Official Review · Reviewer_dNWk · 2022-10-31

**Confidence:** 4
**Correctness:** 2
**Technical Novelty And Significance:** 2
**Empirical Novelty And Significance:** 1
**Recommendation:** 3

**Clarity, Quality, Novelty And Reproducibility:**

**Clarity**

The writing of the paper is unclear and the organization of its content is not that great. The paper is formulation heavy but they are introduces in a unclear/confusing way, e.g., some characters like X and R are used to define various different variables like $X$, $\hat{X}$ and $\bar{X}$, which lack clear explanation and appear to be hard to understand when reading. The organization is not good as well, since many important parts of this work, e.g., literature for AFA methods, math derivations for the method and empirical evaluation results, are presented at appendix. The main paper is not self contained to a great extent and it not fair to let the future audiences to read through the 20+ pages to understand this work.

**Quality**

Overall I feel quality of this paper is limited at the current state because of its developed for solving a small subset of problems in AFAPE, with a very high-level causal graph derivation and weak evaluation result.

n. Also I feel the derived causal graph is too high-level not problem-specific. I do not see apparent components representing the force of *observation missingness* and the authors seem to combine the AFA policy together with environmental factors leading to the *observation missingness* in one variable $R$.

**Novelty**

I feel the novelty is very limited, and the authors overclaim the novelty of various important part of the paper. The authors claim they are the first to introduce the sequence view, but there is a referred work that already done so [Yin et al. 2020]. Also, since the proposed technical contribution of this paper is mostly build on the existing well adopted important sampling regime, the novelty of the solution to their highlighted AFAPE problem is also limited.

**Reproducibility**

The reproducibility of the paper is low. The experiment engages AFA agent trained by two methods, CATE and DQN, implemented on various domains. I do not see sections introducing the important empirical settings, such as hardware/software config, architectural config for the agents, training iterations, etc, on each testified domain if they differ.

**Details Of Ethics Concerns:**

This paper discusses human experiments and it claims its study is related to *safety-critical applications*, e.g., suggesting tests for patient in hospital, but but the declaration on ethics/failure/application on real-world problem is missing.


**Strength And Weaknesses:**

**Strength**

- The problem this paper attempts to tackle is a challenging yet understudied one which is related to many important real-world problem domains such as cost-sensitive diagnosis in healthcare.

**Weaknesses**

[*Scope*] This paper studies the task of evaluating evaluating AFA (Active Feature Acquisition) agents on a very narrow subset of AFA problems where the AFA strategies are defined as reinforcement learning (RL) policies and there is distribution shift between the training and evaluation AFA distributions. I do think the distribution shift for AFA problems is a general challenge existing in both RL-based as well as non RL-based   AFA problems . However, the authors directly connect the AFA problem with off-environment reinforcement learning without claiming the underlying relationship in the two fields, which might mislead the audiences who are new to the AFA literature when reading this paper. Moreover, the main technical solution considered in this work (causal graphs with intervention) is not specifically related to RL. Therefore, only studying performance evaluation (PE) on RL-based AFA problems lacks generality and the work should consider AFAPE with a larger scope.


[*Methodology - Sequence View*]   One major property introduced in the work is about the *sequence view* to select the features, instead of a *set view*.  First, the authors make false claims in several places in the paper, stating the *sequence view* presented in the paper is the *new view on missingness*.  To the best of my knowledge, there are some works from the existing literature studying AFA problems with such *sequence view* already, e.g., [Yin et al, 2020] (cited by the paper). The *sequential view* presented in this work is simply a special case of  the *sequence view* from existing literature. It's unfair to claim the *sequence view* is new when the authors are aware of the work. Second, I do not think the *sequence view* is a better design choice than *set view* for the non-time series AFA problems considered in this paper. For example, when test A and B can be taken in any order, the *sequence view* fail to represent homogeneous property of the sequences $A\rightarrow B$ and $B\rightarrow A$.  Would it be more appropriate to consider an alternative design which combines the *set view* with *sequence view* and let the AFA policy acquire a subset of features at each step in the *sequence view*?

[*Methodology - Observation Missingness*]  (1)  I suggest the authors to rephrase the term. In conventional AFA literature, people refer *observation missingness* as the natural property for any AFA method, since when employing active learning  by default the algorithm would receive partial observation which comes *observation missingness*. I think the *observation missingness* referred in the paper refers to the missingness between train/evaluation stages which is a different type from that in conventional literature. (2) Though generally *observation missingness* could be classified into missing completely at random, missing at random, and missing not at random, it's suspicious if a performance evaluation (PE) method is able to work perfectly well on all the scenarios. Overall I think it's more possible a PE method could work well on a well defined subrange of AFA problems.

[*Methodology - RL perspective*] I think the RL problem considered in the paper is offline RL problem, rather than off-policy RL, because the authors describe the task as off-environment policy training. I think the authors claim their work as off-policy RL mostly in the main paper, even though offline RL has been briefly mentioned from one place in appendix. The authors even claim offline RL is data inefficient, which I think might not make sense. I'm concerned if the RL part of its methodology is correct/clear.

[*Methodology - Causal Graph*] I feel the presented causal graph is too high-level and not significantly novel. It is simplified for some purpose but I expect to see some reasoning over the distribution shift to play around with, such as representing the AFA policy and other factor leading to distributional shift with different variables to derive the formula with. The authors seem to have combined everything leading to *observational missingness* (AFA policy and sources of distributional shift) in one variable $R$, so that it is still unclear what happens to AFA PE model when we deal with the AFA policy and other assumed sources of distributional shift (other than AFA policy) respectively.

[*Evaluation - Choice of baselines*] I feel the choice of  AFA RL policies are insufficient. So far the authors only adopt CATE and DQN, which is insufficient. CATE is an unpublished work and DQN is a weak off-policy baseline. It would be strong if the authors consider state-of-the-art RL methods proposed for AFA problems like SeqVAE.

[*Evaluation - Results*] Though the paper makes rather strong claim saying the method can tackle various of missing patterns, i.e., missing completely at random, missing at random, and missing not at random, the empirical evidence is quite weak.  (1)  the evaluation results in main paper only consists of the MNAR domain, which is insufficient to support the authors' claim. (2) The authors claim their method could obtain the unbiased estimate with Figure 5, but I do not find so. It seems AFAIS with IPW base results in considerable biases compared with the baseline.  (3) There should be some suggestions on which type of methods fit for wha type of missingness, but the authors only present partial view on MNAR.


[*Evaluation - Scenarios*] I think it would be good if the authors could claim their method could work well under various types of interventions which could well cover the observation missingness types. In the current setting, it's unclear how sensitive the proposed PE method would be.

**Other questions**:
- a ML - > an ML
- $\bar{U}^{(k)} = observe(X_i)$: this definition is ambitious as the authors do not introduce the relationship between $i$ and $k$.

**Summary Of The Paper:**

This paper tackle the task of evaluating active feature acquisition (AFA) agent when the AFA agent cannot always access the complete set of features during decision making and there is distribution shift between training and evaluation stages. Specifically, it considers a subset of AFA problems where the features acquired by AFA agent is connected to a classification downstream task and the available dataset contains missing entries that could not be acquired. This work mainly tackles non time series AFA problems where the authors consider the task of feature acquisition from a fixed feature set with a sequence view, for which they claim they are the first one to propose such sequence view.

In the methodology part of this paper, the authors first illustrate the difference between set and sequence views through a hospital example where a task-specific missing data graph (m_graph) is plotted for each view. Then the authors present causal graph for AFA problem, where they identify the feature variable X as the confounder because they claim there exits a path $\bar{R} \leftarrow X \rightarrow C$. They also declare that AFA agent is allowed to implement causal interventions on $\bar{R}$ so that the problem of AFA PE can be formulated as a task of evaluating the cost objective $\mathcal{J}$ under causal intervention on $\bar{R}$. To correct the distributional shift for training and testing stages, the authors propose to integrate an existing technique for unbiased estimation, i.e., importance sampling, to their performance evaluation method.

The authors make strong claim that their method could work on various type of missing observation patterns, including missing completely at random (MCAR), missing at random (MAR), and missing not at random (MNAR). In the main paper, they show empirical results only for MNAR.

**Summary Of The Review:**

This paper learns an interesting problem of AFAPE, but I have concerns on its writing, problem scope, the capability of tackling all different types of *observational missingness* in one causal graph, the over simplified causal graph and its insufficient empirical evaluation.  There's a possibility the proposed method could work well on a well specified subset of AFA problems with certain property.

---

> ### Author Response · Authors · 2022-11-19
> **Response to Weakness 1 and 2**
>
> We thank the reviewer for their valuable time reviewing and thoughtful feedback.
>
> Weakness 1: Scope
>
> 1) Narrow subset of AFA problems:
>
> The reviewer mentions the paper studies a very narrow subset of AFA problems where the AFA strategies are defined as RL policies. The reviewer, however, also mentions that our solution is more general and not specifically related to RL-based policies. The latter is correct. We also have a sentence stating this under related methods: "This work focuses, however, not on any particular AFA
> method, but on the evaluation of any AFA method under missingness". The wording is, however, RL heavy and we thank the reviewer to point out that one could be misguided to think that only RL policies could be evaluated.
>
> Our approach can thus cover a large subset of AFA problems. Admittedly, we do only consider classification as the prediction task of interest, but the type of prediction task plays only a negligible role for the evaluation step and extensions are therefore straight-forward. Furthermore, we only consider the static setting and no time-series settings. This definitely is a strong limitation and we consider an extension to time-series settings as an extremely important step required for a wide range of practical applications. Time-series AFAPE problems, however, have additional challenges and we thus leave them for future work.
>
> Action:
> We generalized wording with respect to RL where appropriate to clarify that any AFA policy can be evaluated.
>
> 2) Off-environment RL and AFA:
>
> The reviewer is concerned about a lack of proof for the relationship between RL and AFA. In section Section 3.2 "PROBLEM FORMULATION: ACTIVE FEATURE ACQUISITION (AFA)" we show rigorously how the AFA problem can be formulated as a sequential decision process which establishes the relationship between AFA and RL. The off-environment view is subsequently derived based on the AFA graph. The formulation as a sequential decision process is the most fine-grained formulation possible for AFA as it allows the new information of each measured feature to be used by the agent. Any AFA method that does not take this view either considers further constraints on the problem that we disregard here (e.g. that one cannot always wait for results before making new observation decisions) or is a simplified approach of this general formulation.
>
> Weakness 2: Sequence view
>
> 1) Sequence view as a novelty
>
> The reviewer is right in that we are not the first to represent the AFA problem as a sequential decision process (which requires the sequence view). It is a necessary step to be taken in every RL-based AFA paper. The wording of the novelty was thus unfortunate and we have removed this claim. The intent of our claim was to point out that we are the first to relate the two views on missingness (set and sequence view) and therefore establish the relationship to the well-established missing data literature. This allows the discussion and comparison of different estimators that are based on different views on missingness.
>
> Action: We have removed claims about the sequence view being a novelty.
>
> 2) Sequence view vs set view
>
> The reviewer further mentions concerns the sequence view might not be the best way of looking at the problem. We want to clarify that i) there are two types of missingness (observational/retrospective and AFA missingness) and for each we must decide which of the two views to take; ii) our discussion does not address how the agent should "internally" encode the data (we recommend to use a state space $\bar{X}^{(k)}, \bar{R}^{(k)}$, i.e. a set view that makes use of the mentioned homogeneous property); and iii) We propose to change views based on the task:
>
> 1) For the step of defining an AFA policy (and training it) we must choose the sequence view for AFA missingness. The order plays a big role (after observing $A$ we might decide not to observe $B$ after all). Acquiring a subset of features at each step, as proposed by the reviewer, leads to suboptimal decisions, because by observing e.g. $A$ and $B$ jointly, the agent is stripped from the opportunity to decide that $B$ might not be necessary after viewing $A$.
>
> 2) For the evaluation step, we absolutely agree with the reviewer that the set view (for AFA missingness) is more suitable in order to make use of the mentioned homogenous property. This is exactly what we do when employing the off-environment bandit view. The off-environment bandit view looks at both retrospective/observational missingness and AFA missingness from a set view.
>
> Action: We added an additional Appendix (Appendix A.10) that derives the data efficiencies for the different views. This hopefully helps to prevent confusions about the switching of views and clarifies the benefit of the off-environment bandit view.

---

> > ### Author Response · Authors · 2022-11-19
> > **Response to Weakness 3 and 4**
> >
> > Weakness 3: Observation Missingness
> >
> > 1) Other word for observational missingness
> >
> > Thank you for pointing out this possible source of confusion.
> >
> > Action: We have changed the term to retrospective missingness.
> >
> > 2) MCAR, MAR and MNAR
> >
> > The reviewer also mentions their concern that a performance evaluation (PE) method is able to work perfectly well on all (MCAR, MAR and MNAR) missing data scenarios. In general, MNAR missingness scenarios are marked by their identifiability property, i.e. if the target distribution is identified under the assumed model. Our approach is no exception to this fact. It is a general approach to estimation, not a substitute for the identification step.
> >
> > For problems that are in general not identified or for which the AFA propensity score cannot be identified, AFAIS will therefore not be able to produce unbiased estimates. In fact, the notion of estimation (unbiased or otherwise) is not well defined in non-identified problems.
> > We also mention that we do not give general identifiability results for MNAR missingness scenarios (in Section 3.4.3). If identification is, however, possible, we show that AFAIS is unbiased and consistent!
> > Note that the calculation of the weight within AFAIS will depend on what precise function of the observed data the weight is, as dictated by identification theory for MNAR models.
> >
> > Action: We have rephrased the description of our experiments in multiple parts of the paper to make sure the reader understands that we only test examples of MCAR, MAR and MNAR missingness (as it is infeasible to test all different patterns of MNAR that are possible). For example in Section 4.1 we write: "We evaluate different estimators on synthetic and real world datasets under examples of synthetic MCAR, MAR and MNAR retrospective missingness." We further extended the Discussion section to emphasize the importance of the identification step.
> >
> > Weakness 4: RL perspective
> >
> > 1) Off-policy policy evaluation (OPE):
> >
> > OPE describes the task of evaluating a target policy from data created by a behavior policy. This is a well established field and describes precisely the approach we discuss in Appendix A.9. This approach is taken if one views both observational/retrospective missingness and AFA missingness from a sequence viewpoint. It is, however, not the viewpoint we recommend. We show in this paper that AFAPE is not only an OPE problem, nor only a missing data problem, but that there are multiple views on the problem with different advantages /disadvantages and that we recommend the off-environment bandit view.
> > We chose to use the term "OPE" (instead of offline RL) to lead the reader directly to the relevant literature and emphasize that we look at policy evaluation and not learning. When an OPE perspective is taken, one views the retrospective missingness policy as the behavior policy. This policy cannot be changed and one can thus classify the learning problem as offline RL.
> >
> > Action: We added a sentence in Section 2 (Related Methods) to clarify the relationship to offline RL:
> > "The goal in OPE is to evaluate the performance  of a "target" policy (here the AFA policy) from data collected under a "behavior" policy (here the retrospective missingness induced by e.g. the doctor). In this special case, the behavior policy cannot be changed, leading to offline reinforcement learning".
> >
> > 2) Inefficient offline RL:
> >
> > The reviewer also mentions our claim that offline RL is data inefficient in AFA settings does not make sense. We justified this claim in Appendix A.9, but hadn't quantified the data inefficiency. We have now added an Appendix (Appendix A.10) in which we derive the data efficiency of the different views showing that the OPE view is extremely data inefficient. Our derivations show that under the OPE view (RL perspective), one trajectory of actions can be evaluated from one data point. Under the off-environment bandit view, one can evaluate $\sum_{i=0}^{\lVert \bar{R} \rVert_{1}}i! {{\lVert \bar{R} \rVert_{1}} \choose i}$ different trajectories from one data point (where ${\lVert \bar{R} \rVert_{1}}$ denotes the number of available features for that data point). Imagine a fully observed data point with 10 features. It can be used to evaluate 10 million different trajectories in the off-environment bandit view and one  trajectory in the OPE view.
> >
> > Action: We added Appendix A.10 to quantify data efficiency.

---

> > > ### Author Response · Authors · 2022-11-19
> > > **Response to Weakness 5,6 and 7**
> > >
> > > Weakness 5: Causal graph
> > >
> > > We believe the AFA graph is the central element of our approach and therefore the core of the novelty. It is only simplified in the sense that the underlying m-graph is not shown in the graph explicitly but hidden in the relationship $X \rightarrow \bar{R}$. This was chosen to make the graph easily understandable and as general as possible (it holds for any missingness scenario). The graph contains all relevant variables to derive the formulas for AFAIS. It further allows the interpretation of the AFAPE problem as an off-environment bandit problem.
> > >
> > > We would like to comment on the following sentence in particular: "The authors seem to have combined everything leading to observational missingness (AFA policy and sources of distributional shift) in one variable $R$, so that it is still unclear what happens to AFA PE model when we deal with the AFA policy and other assumed sources of distributional shift (other than AFA policy) respectively."
> > >
> > > We denote the AFA policy as the policy we try to learn (i.e. the "AI" policy). We denote the policy that created the data (i.e. the doctors decisions) as the observational/retrospective missingness policy. Therefore, the AFA policy does not lead to observational/retrospective missingness (the AI method we design cannot influence the retrospective dataset).
> > > Additionally, there are two variables $\bar{R}$ and $\hat{R}$ representing observational/retrospective and AFA missingness, respectively. We consider distribution shift only in the retrospective missingness indicator $\bar{R}$.
> > > From the causal inference perspective, the distribution shift is modelled not by a distinct variable (node) but by a structural change to the DAG, namely removing of causal edges. To comply with this perspective, we have modelled the distributional shift by removing the edge $X \rightarrow \bar{R}$ (corresponding to the intervention $do(\bar{R}=\vec{1})$). Note that the complexity of the distribution shift is thus hidden in the relationship $X \rightarrow \bar{R}$ which can be represented separately by a graphical model, the m-graph.
> > >
> > > Weakness 6: Choice of baselines
> > >
> > > We agree with the reviewers that there are more novel AFA approaches than the tested baselines. The goal of this paper is, however, not to test different AFA methods, but to compare the performance of estimators. The choice of AFA agent we evaluate is thus not important to benchmark estimators.
> > >
> > > As a clarifying analogy, our approach to AFA is what performance metrics are to a classification problem. Regardless of the classifier, the metric must reflect the true classification. Similarly, our method provides unbiased performance evaluation estimates regardless of the AFA agent.
> > >
> > > Action: We removed CATE from the experiments.
> > >
> > > Weakness 7: Results
> > >
> > > 1) Only MNAR results in the main paper:
> > >
> > > We restructured the paper to also include MCAR and MAR results in the main paper.
> > > The shown MNAR model makes strictly stronger assumptions than the MCAR and MAR missingness patterns. Showing that an estimator works on the MNAR data thus also shows that it would work on MCAR or MAR data.
> > >
> > > Action: We moved the real-world data results for MCAR, MAR and MNAR from the Appendix to the main paper.
> > >
> > > 2) Unbiased estimate in Figure 5:
> > >
> > > We disagree with the reviewers concern that Figure 5 would not show unbiasedness of the IPW and AFAIS estimators. The estimation error (y-axis) clearly decreases as the dataset size increases (y-axis has log-scale!). Obviously the error does not shrink as fast as if there was no missingness (ground truth), but this does not mean the results show bias (only that the convergence is slower than if we had fully observed data). Please also have a look at Figure A.5 that shows the actual estimates for the same experiment.
> > >
> > > 3) Which method for which missingness:
> > >
> > > Thank you for this suggestion. We added to the Discussion to clarify the main questions that should be answered when deciding on an estimator. These include identification, data efficiency and estimation of the nuisance functions (propensity and AFA propensity scores).
> > > Regarding identification, we also mention in Sections 3.4.2 and 3.4.3 the problems of identification of AFAIS and discuss which method should be used depending on the missingness mechanism (i.e. moving on the AFAIS-IPW spectrum). We also give a guide on how to choose $s$ (in Appendix A.8).
> > >
> > > Action: We added to the Discussion to clarify which problem characteristics should influence the choice of the estimator.

---

> > > > ### Author Response · Authors · 2022-11-19
> > > > **Response to Weakness 8, other questions, "Clarity, Quality, Novelty And Reproducibility" and ethical concerns**
> > > >
> > > > Weakness 8: Scenarios
> > > >
> > > > We are not sure what this recommendation is addressing. We have precisely derived the estimator for the intervention $do(\bar{R}= \vec{1})$ which corresponds to an "availability" distribution shift from "only observed features in the retrospective dataset are available" to "all features are available". We do not consider other types of interventions and the proposed framework will not be able to correct for those.
> > > >
> > > > Other questions:
> > > >
> > > > Thank you for pointing these out.
> > > >
> > > > Action: Changed sentence to: "We assume that a missingness indicator $\bar{R}$ is formed during a sequence of observation actions ${\bar{U} = (\bar{U}^{(1)},...,\bar{U}^{(N)})}$, where action $\bar{U}^{(k)}$ can take on values in \{"observe($X_1$)","observe($X_2$)",...\} if $k<N$."
> > > >
> > > > Clarity:
> > > >
> > > > 1) Lack of clarity regarding missingness notation
> > > >
> > > > Thank you for pointing this out. We've made some changes to help the reader follow the notation.
> > > >
> > > > Action:
> > > > We added in section 3.3, when introducing retrospective missingness, a sentence to clarify the notation: "Note that, in contrast to the AFA missingness (denoted by $\hat{R}$, $\hat{U}$  and $\hat{X}$), the retrospective missingness (denoted by $\bar{R}$, $\bar{U}$ and $\bar{X}$) is not necessarily restricted to $\mathcal{F}_\textit{MAR}$."
> > > >
> > > > We have also included a reference to the glossary in the main text (beginning of Section 3.1).
> > > >
> > > > 1) Paper structure
> > > >
> > > > We would have liked to include important parts of the work such as AFA method literature review, derivations and results in the main body.
> > > > Since the demand for more empirical results in the main body was raised by several reviewers we have changed the structure now to include additional results.
> > > > Due to the page limit of 9 pages, it is, however, unfortunately not possible to include all of the mentioned sections in the main body. Additionally, there is a conflicting request by another reviewer of reducing the Methods section (instead of adding derivations).
> > > > As the derivations can take away from the readability and might be only of interest for some parts of the audience, we continue to keep them in the Appendix. Regarding the literature review on AFA methods, we believe, as we do not propose a new AFA method, it is more beneficial to use the space in the main body for other parts.
> > > >
> > > > Action: We've added empirical results to the main body and extended the Discussion section.
> > > >
> > > > Quality:
> > > >
> > > > We hope our arguments have resolved the reviewers concerns about the small subset of problems in AFAPE (we can evaluate non RL-based and RL-based), the high-level causal graph (complicated m-graph is problem specific and hidden in the AFA graph and the distribution shift is modelled as a structural change in the graph) and the evaluation results (we added experiment results from appendix to main body).
> > > >
> > > > Novelty:
> > > >
> > > > We hope our correction of not claiming the sequence view as our own contribution has erased the reviewers concerns of overstated claims. Furthermore, we would like to point out that importance sampling is a direct result of Bayes rule. What is demanding about importance sampling and often calls for non-trivial novelties is the selection of the sampling distribution, which directly affects the veracity and practicality of the proposed solution. Our newly added quantification of data efficiency demonstrates this clearly (Appendix A.10). While all views (OPE, missing data, off-environment bandit) have IS-based estimators, their data efficiency can differ tremendously (several orders of magnitude, as our example in the answer to weakness 4 shows).
> > > >
> > > > Reproducability:
> > > >
> > > > We agree with the reviewer about a previous lack of reproducability.
> > > >
> > > > Action:
> > > > We have added extensive experiment details in Appendix A.11 and hope these can resolve the raised concerns about reproducability.
> > > >
> > > > Details of Ethics Concerns:
> > > >
> > > > We did not conduct any data collection and all real-world datasets used in this work are publicly available on recognized websites.
> > > > Therefore, our understanding is that we don't need to to provide an ethics statement, but we are happy to follow ICLR guidelines.

---

> > > > > ### Author Response · Authors · 2022-12-02
> > > > > **Response to Reviewer**
> > > > >
> > > > > We would like to kindly thank you again for your comments and feedback. In light of the upcoming discussion deadline, we would appreciate it very much if you could let us know whether all your questions are addressed. We are also more than happy to answer any further questions.

---

### Decision · Program_Chairs · 2023-01-20

**Decision:**

Reject

**Justification For Why Not Higher Score:**

As said, the paper addresses an important problem and in general, has interesting contributions. So the paper could be accepted - but a future version of the paper is likely much better.

**Justification For Why Not Lower Score:**

n/a

**Metareview: Summary, Strengths And Weaknesses:**

This paper considers the problem of performance evaluation for active feature acquisition (AFA). In particular, they highlight the distribution shift that typically arises in this setting and how unbiased estimation can be performed. In particular, they consider an importance sampling-based estimator (inverse probability weighting, IPW) but observe that it might be very data efficient. To this end, they propose a novel approach, AFAIS (active feature acquisition importance sampling), exploiting the particular properties of the AFA problem. They validate their approach and several baselines in synthetic and semi-real-world experiments, illustrating that their approach can perform better than the baselines.

There was no author-reviewer discussion on OpenReview unfortunately but the reviewers considered the authors' rebuttal in our virtual discussion.

Strengths of the paper.
The paper addresses an important problem that is sometimes incorrectly approached in the literature and the careful causal analysis sheds light on how to do a better job (in some cases). IPW is a standard technique but in contrast to most literature, they apply it properly to the AFA setting. AFAIS is novel and more efficient than the baselines in some cases.

Weaknesses of the paper.
The biggest weakness of the paper is its presentation/clarity and limited experiments / limited insights from experiments. All reviewers shared concerns in that regard. In particular, the authors should make all necessary assumptions explicit (in particular regarding their experimental findings) and also concisely explain their theoretical results in the main paper (e.g. regarding the unbiasedness and consistency of AFAIS). More relevant experiments should be presented in the main body of the paper, also regarding the limits of the method (which missingness cases work, etc.). More generally, the authors should consider stronger real-world-inspired experiments regarding missingness. Also, AFAIS cannot be applied in all cases - the main paper should clearly detail what AFAIS can do and what it cannot do.

In the virtual discussion, we came to the consensus that the presentation of the paper must be improved before publication. But we also want to emphasize that the paper is really interesting and will be of great interest to the community. So the authors are encouraged to improve their paper in line with the reviews and submit their work to an upcoming venue.

**Summary Of Ac-Reviewer Meeting:**

Reviewers dNWk, qb3R and zfWY took part in the virtual discussion. Reviewers Pwju and Vxt6 could not attend the meeting.

The reviewers stated that they found the problem addressed in the paper important and the contributions are in general relevant, but that experiments are weak and that the presentation of the paper is unclear and unsatisfying in many parts. The rebuttal clarified some unclarities but it did not resolve the concerns regarding experiments and presentation.

Hence the discussion revolved around how to weigh these points against each other. We came to the consensus that the paper should not be accepted in its current form but we'd like to encourage the authors to improve their paper according to the reviews and submit it to an upcoming venue.